# Aurora-A mediated histone H3 phosphorylation of threonine 118 controls condensin I and cohesin occupancy in mitosis

Candice L Wike[1], Hillary K Graves[1], Reva Hawkins[1], Matthew D Gibson[2], Michelle B Ferdinand[3], Tao Zhang[4], Zhihong Chen[1], Damien F Hudson[4], Jennifer J Ottesen[3], Michael G Poirier[2], Jill Schumacher[5], Jessica K Tyler[1]*†

[1]Department of Epigenetics and Molecular Carcinogenesis, University of Texas MD Anderson Cancer Center, Houston, United States; [2]Department of Physics, The Ohio State University, Columbus, United States; [3]Department of Chemistry and Biochemistry, The Ohio State University, Columbus, United States; [4]Murdoch Children's Research Institute, Royal Children's Hospital, Melbourne, Australia; [5]Department of Genetics, University of Texas MD Anderson Cancer Center, Houston, United States

*For correspondence: jet2021@med.cornell.edu

Present address: †Department of Pathology and Laboratory Medicine, Weill Cornell Medicine, New York, United States

**Abstract** Phosphorylation of histone H3 threonine 118 (H3 T118ph) weakens histone DNA-contacts, disrupting the nucleosome structure. We show that Aurora-A mediated H3 T118ph occurs at pericentromeres and chromosome arms during prophase and is lost upon chromosome alignment. Expression of H3 T118E or H3 T118I (a SIN mutation that bypasses the need for the ATP-dependent nucleosome remodeler SWI/SNF) leads to mitotic problems including defects in spindle attachment, delayed cytokinesis, reduced chromatin packaging, cohesion loss, cohesin and condensin I loss in human cells. In agreement, overexpression of Aurora-A leads to increased H3 T118ph levels, causing cohesion loss, and reduced levels of cohesin and condensin I on chromatin. Normal levels of H3 T118ph are important because it is required for development in fruit flies. We propose that H3 T118ph alters the chromatin structure during specific phases of mitosis to promote timely condensin I and cohesin disassociation, which is essential for effective chromosome segregation.

## Introduction

The packaging of the eukaryotic genome into chromatin facilitates the temporal and spatial regulation of all genomic activities, including DNA repair, replication, transcription and mitosis. Chromatin comprises arrays of nucleosomes, where each nucleosome has ~146 base pairs of DNA wrapped 1.75 times around a histone octamer composed of two molecules each of core histone H3, H4, H2A, and H2B (*Kornberg, 1974*). Repetitive arrays of nucleosomes are then further compacted by higher-order folding, requiring additional proteins including linker histones. During mitosis, chromosome condensation plays a critical role in preventing DNA breaks during mitosis and enabling equal chromosome segregation to the two daughter cells (*Ganem and Pellman, 2012*).

One important means by which the cell achieves accurate regulation of genomic processes, including mitosis, is via post-translational modifications (PTMs) of the core histones (*Strahl and Allis, 2000*). The PTMs, usually occurring on the N- and C-terminal tails of the histones, generally serve to recruit reader proteins to the chromatin. PTMs also occur on the histone globular domains, but are

**eLife digest** In every one of our cells, our DNA is wrapped together with histone proteins to make a structure called chromatin. When a cell divides, each newly formed daughter cell must receive an identical set of chromatin. As part of this process, the chromatin is copied and then compacted, which causes a characteristic "X"-shaped chromosome to form. This "X" shape is actually made up of two identical parts, or chromatids, that are joined together until a specific time during cell division. If chromosomes separate too early or too late, the DNA will not distribute evenly to daughter cells, which could lead to diseases including cancer.

Histone modifications are small chemical changes on the histone proteins that the DNA wraps around. Previous research identified a new histone modification that is located at an important contact point between the DNA and a particular histone protein. However, the role of this modification in living cells was not clear.

Wike et al. have now determined that in animal cells, this histone modification occurs immediately before the chromatids separate and at specific locations along the chromosomes. The amount of this histone modification is very important: in cells with too much of the modification, the chromosomes compacted incorrectly and the chromatids separated too soon. As a result, the chromosomes were incorrectly distributed among the daughter cells.

Wike et al. also show that an enzyme called Aurora-A kinase is responsible for making this histone modification. The next challenge will be to understand how the Aurora-A kinase knows when and where to add the histone modification to the chromosome. This will help us to understand how the overproduction of Aurora-A causes cancer.

much less well studied than the histone tail modifications. PTMs at the histone-DNA interface have been proposed to directly modulate nucleosome structure, without the need for reader proteins (*Cosgrove et al., 2004*). Of all the histone PTMs that occur at the histone-DNA interface, one of the best positioned to disrupt the nucleosome structure is phosphorylation of threonine 118 (T118ph) of H3 (*Mersfelder and Parthun, 2006*). In agreement with its important location within the nucleosome structure (*Figure 1A*), biochemical studies have confirmed that H3 T118ph causes reduced nucleosome stability, increased nucleosome mobility, and increased DNA accessibility (*North et al., 2011*). Strikingly, H3 T118ph caused the formation of novel populations of alternate DNA-histone complexes that have DNA wrapped around two complete histone octamers arranged edge-to-edge, termed nucleosome duplexes and altosomes (*North et al., 2014*). In agreement with the biochemical data, a substitution of H3 T118 for isoleucine (T118I) was identified in *S. cerevisiae* as a dominant Swi-INdependent (SIN) (*Kruger et al., 1995*). The SIN H3 T118I substitution allows nucleosomes to slide along the DNA without the need for SWI/SNF (*Muthurajan et al., 2004*).

Despite the striking biochemical effects of H3 T118ph on nucleosome structure and the phenotype of the yeast T118I mutant, H3 T118ph has not been studied in cells beyond its identification (*Olsen et al., 2010*). Accordingly, we characterized H3 T118ph function in metazoan cells. H3 T118ph, mediated by Aurora-A, is localized to centromeres and chromosome arms during specific phases of mitosis, Mutation of H3 T118 caused a wealth of defects including lagging chromosomes, delayed cytokinesis, reduced cohesion and altered chromosome compaction in mammalian cells and inviability in *Drosophila*. Given that the H3 T118I mutant or overexpression of Aurora-A led to premature release of cohesin and condensin I from chromosomes, we propose that H3 T118ph alters chromosome structure during mitosis to help dissociate cohesion and condensin I.

## Results

### H3 T118ph is dynamically regulated during mitosis in metazoans

To characterize the spatiotemporal localization of H3 T118ph (*Figure 1A*), we first established the specificity of the H3 T118ph antibodies. Here we show only the results obtained with the Abcam antibody, although similar results were obtained with our independently generated H3 T118ph polyclonal antisera (data not shown). The antisera were highly specific in dot-blot assays (*Figure 1B*) and

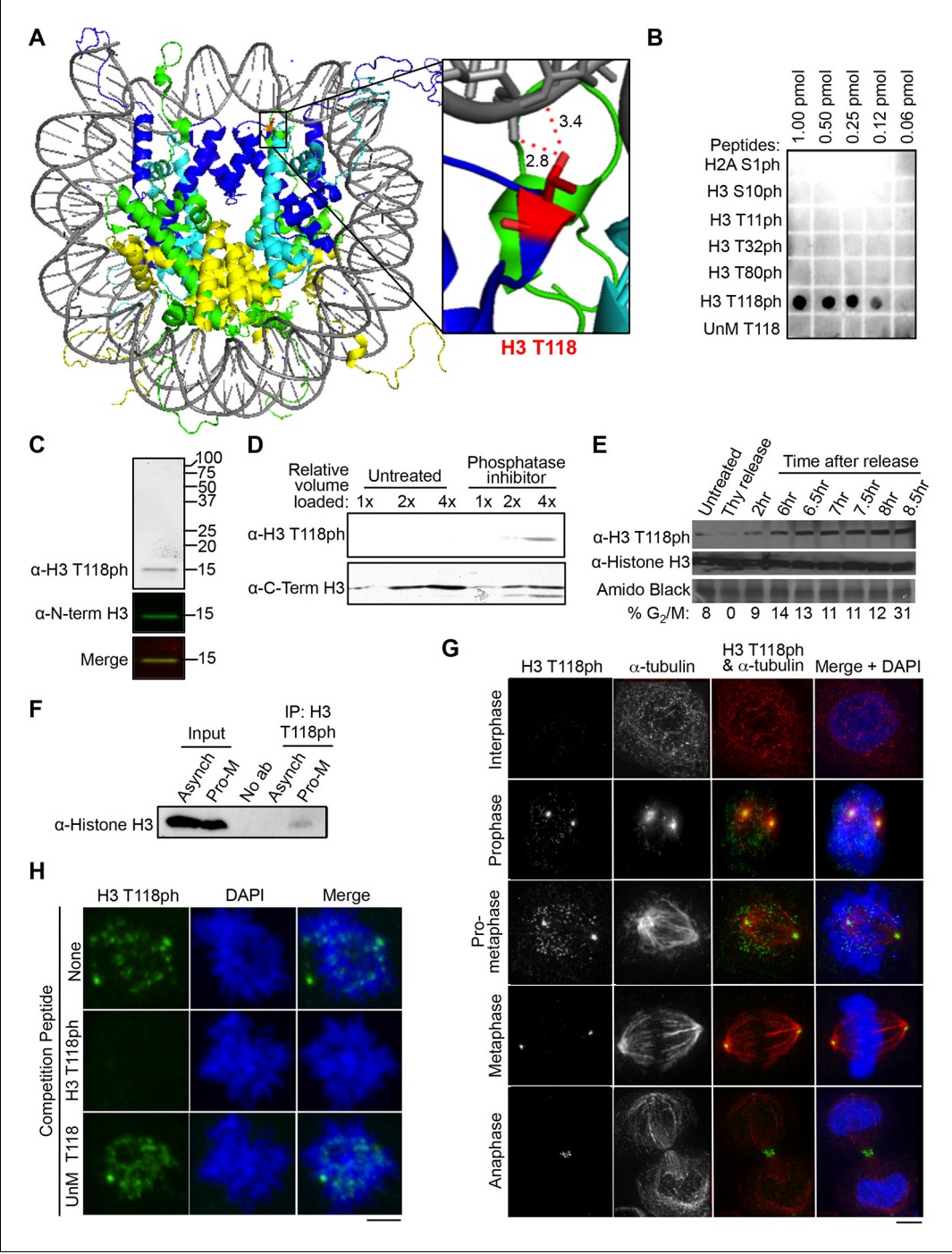

**Figure 1.** Dynamic mitotic phosphorylation of H3 T118. (**A**) The side chain of H3 T118 (red) is close enough to form a hydrogen bond with the DNA (grey). Histone H3 is depicted in dark blue, Histone H4 is cyan, Histone H2A is green and H2B is yellow. Angstrom distances were drawn using nearest neighbor wizard in pymol. Protein Data Bank (PDB) code 1KX5. (**B**) The indicated amounts of the respective peptides were dotted and the membrane probed with an antibody to histone H3 T118ph. The UnM T118 peptide corresponds to human histone H3 aa 115 to 125. (**C**) Western blot of crude extract from HeLa cells, using infra-red labeled secondary antibodies. H3 T118ph (greyscale/red) and N-term histone H3 (green). (**D**) HeLa cell extracts untreated or treated with phosphatase inhibitor were probed with the indicated antibodies. Full western blot image can be found in *Figure 1—figure supplement 1A*. (**E**) HeLa cells were synchronized by a double thymidine arrest and released at the indicated times, followed by western blot analysis of whole cell extracts. (**F**) Immunoprecipitation (IP) using the H3 T118ph antibody from HeLa cells asynchronous (Asynch) or released from a $G_2$ arrest (with 9 µM Ro-3306 for 16 hr) for 30

*Figure 1 continued on next page*

*Figure 1 continued*

min resulting in pro-metaphase cells (Pro-M). Full western blot image can be found in *Figure 1—figure supplement 1B*. (G) Immunofluorescence analysis of H3 T118ph (green) and α-tubulin (red) in HeLa cells. Scale bar = 5 μm. (H) H3 T118ph antibody was pre-incubated with no peptide (top), H3 phosphorylated at T118 (middle) or unmodified (UnM T118, bottom). The supernatants were used to detect H3 T118ph in pro-metaphase HeLa cells. Scale bar = 5 μm.

The following figure supplement is available for figure 1:

**Figure supplement 1.** Full size western blots of data shown in *Figure 1*.

---

recognize a single protein identical in size to histone H3 in western blot analysis of total protein extracts from HeLa cells (*Figure 1C*). This signal in western blots was increased by treating the cells with the protein phosphatase 1 and 2A inhibitor calyculin A for 3 hr, indicating that the H3 T118ph antibody recognized phosphorylated H3 (*Figure 1D*, *Figure 1—figure supplement 1A*). In concordance with previously published mass spectrometry results (*Olsen et al., 2010*), we observed a dramatic increase in H3 T118ph levels as cells entered mitosis (*Figure 1E*). The antibody also recognized H3 T118ph in its native conformation, because it immunoprecipitated H3 from cells released into mitosis (*Figure 1F*, *Figure 1—figure supplement 1B*). Using immunofluorescence analysis, we found that H3 T118ph was restricted to mitotic cells during prophase through anaphase and was greatly diminished in interphase (*Figure 1G*). Specifically, H3 T118ph signal was detected as discrete foci on chromatin only in prophase and pro-metaphase. Additionally, H3 T118ph co-localized with centrosomes through all phases of mitosis (*Figure 1G*). This is a consequence of non-chromatin bound histones localizing to the centrosomes for proteasome-mediated degradation during mitosis (C. Wike and J.K. Tyler, manuscript submitted). During anaphase, the H3 T118ph antibodies also detected the spindle mid-body (*Figure 1G*). The localization pattern of H3 T118ph was not unique to HeLa cells, nor cancer cell lines, because it was similar in HMEC, WI-38 and MCF10A cells (data not shown). Finally, the H3 T118ph signal was specifically competed away by an H3 T118ph peptide (*Figure 1H*). Together, these results show that the H3 T118ph antibody is specific.

## H3 T118ph localizes to centromeres and chromosome arms during prophase and pro-metaphase

Threonine 118 and the surrounding residues are highly conserved among metazoan H3 proteins. Therefore, we tested whether H3 T118 is phosphorylated in other metazoans and whether this occurs specifically during mitosis. In *D. melanogaster* S2 cells, H3 T118ph localized to chromatin and centrosomes during mitosis (data not shown). H3 T118ph localization was also conserved in *C. elegans*. During pro-metaphase, H3 T118ph was localized along the outside edges of chromosomes, indicative of centromeric localization on holocentric chromosomes in *C. elegans* (*Figure 2A*). To determine if the localization of H3 T118ph along the arms of chromosomes was dependent on the centromeric chromatin structure, we used siRNA to the centromeric histone variant CENP-A to abolish the centromeres. Upon CENP-A knockdown, H3 T118ph is diminished from the chromatin (*Figure 2A*). These data demonstrate that mitotic enrichment of H3 T118ph is conserved amongst metazoans. Furthermore, H3 T118ph localizes to centromeres and its localization is dependent on intact centromeres.

Given our results in *C. elegans*, we asked if the punctate chromosomal staining of H3 T118ph in mammalian cells (*Figure 1G*) reflects centromeric staining. Indeed, we found that H3 T118ph co-localized with CENP-A in human cells in prophase and pro-metaphase (*Figure 2B*). Noteworthy, the prophase to metaphase timing of the appearance and disappearance of H3 T118ph on centromeres is distinct from other mitotic H3 phosphorylation events. For example, H3 S10ph (*Crosio et al., 2002*) and H3 T3ph (*Polioudaki et al., 2004*) remain on chromosome arms and centromeres, respectively, through anaphase. CENP-A S7ph (*Zeitlin et al., 2001*) remains through metaphase, while H3 T118ph is lost from centromeres in metaphase coincident with chromosome alignment. H3 T118ph foci did not always perfectly colocalize with CENP-A, but sometimes appeared to be adjacent to CENP-A foci. Indeed, detailed inspection of mitotic spreads revealed that H3 T118ph localized to the inner centromere when cells were treated with the microtubule destabilizing drug colcemid while

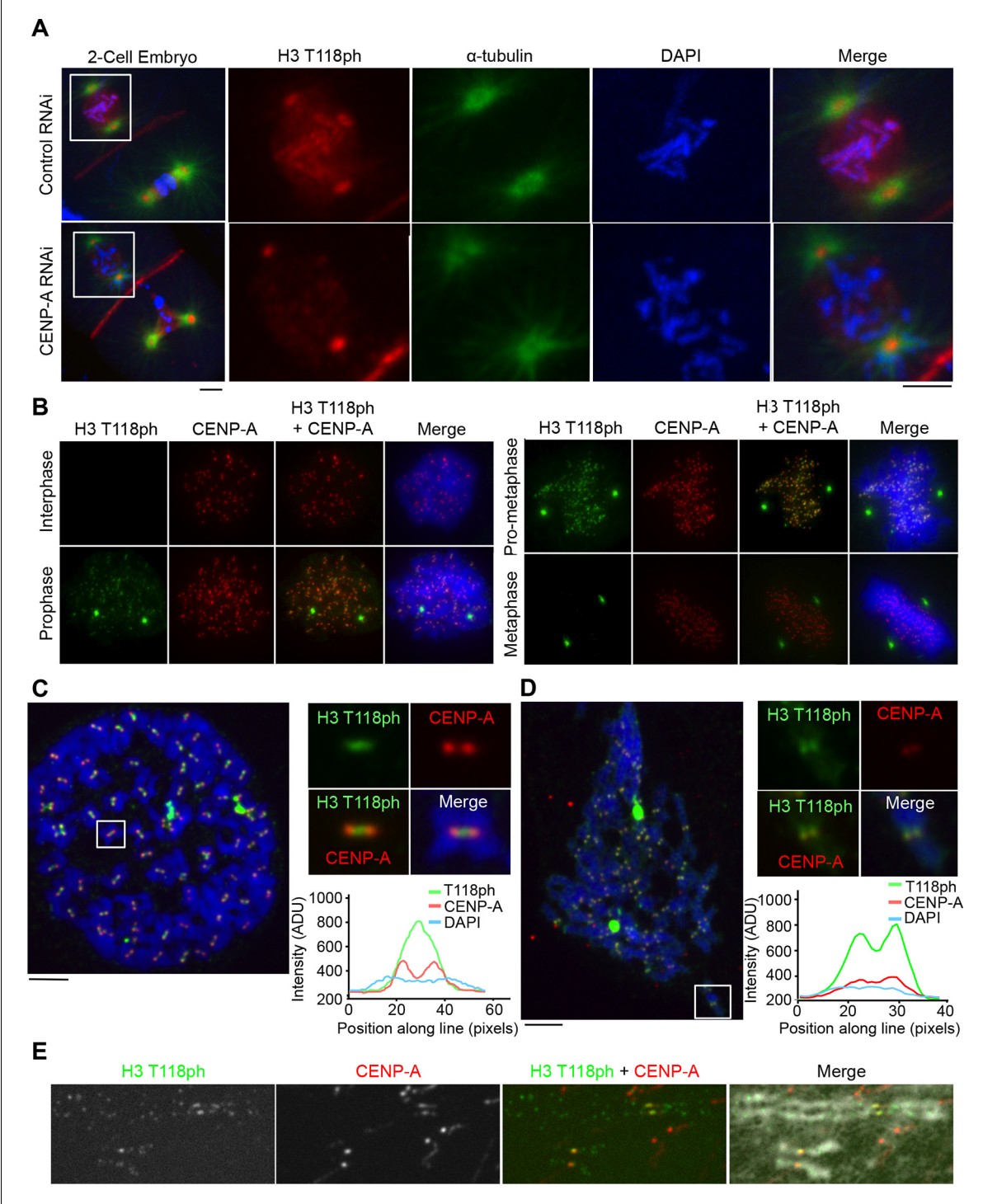

**Figure 2.** H3 T118ph localizes to pericentromeres and chromosome arms during prophase and pro-metaphase. (**A**) Immunofluorescence of two-cell *C. elegans* embryos Control (RNAi) (top) and centromeric protein A CENP-A (RNAi)-depleted (bottom) embryos were fixed and stained with α-tubulin (green) and H3 T118ph (red) antibodies. DNA was stained with DAPI (blue). Scale bar = 5 μm. (**B-E**) Immunofluorescence of HeLa cells stained with CENP-A (red) and H3 T118ph (green) antibodies. (**B**) Images of progressive mitotic stages. (**C**) Mitotic spreads synchronized with colcemid (no tension across the kinetochores). The white box indicates magnified area. Intensity of the signal across centromeres is plotted. Scale bar = 5 μm. (**D**) Unsynchronized mitotic spread, as in C. (**E**) Extended metaphase chromatid fibers showing H3 T118ph localization to discrete regions of chromosome arms.

CENP-A remained on the outer-centromere (*Figure 2C*). The distinct localization of CENP-A and H3 T118ph emphasizes that the H3 T118ph signal is not due to phosphorylation of CENP-A per se. Importantly, upon colcemid treatment, where microtubule attachment is lost, the interkinetochore distance is decreased (*Uchida et al., 2009*). The single foci that is detected by H3 T118ph antibody could be because of a single population of H3 T118ph localized to the inner-centromere or because the two adjacent centromeres are separated by less than the resolution of light microscopy, which is theoretically 200nm. Therefore we asked when there is dynamic microtubule attachment, promoting tension and kinetochore stretching, what is the true H3 T118ph signal at the centromere. H3 T118ph existed in two foci per chromosome that correlate well with, but are larger than, the two CENP-A foci (*Figure 2D*) consistent with pericentromeric localization. We noted that H3 T118ph also appears to occur along the chromosome arms in mitotic spreads (*Figure 2D*). Indeed, H3 T118ph was detectable at the centromere partially overlapping with CENP-A and at weaker foci at discrete intervals along the chromosome arms on extended chromatin fibers (*Figure 2E*).

## Aurora-A phosphorylates H3 T118

To gain insight into the function of H3 T118ph, we sought to identify the kinase responsible for its phosphorylation. We utilized a ProQinase kinase screen to test 190 recombinant kinases for their ability to phosphorylate H3 T118 (*Figure 3—figure supplement 1*). Aurora-A was the only cell cycle regulated kinase able to efficiently phosphorylate H3 T118 from among the three positive kinases, arbitrarily defined as having an activity above 3000 cpm. Absence of phosphorylation of H3 T118 by Aurora-B and Aurora-C further validated Aurora-A as the kinase of H3 T118 (*Figure 3A*). We independently confirmed that Aurora-B INCENP could not phosphorylate the T118 peptide, including an H3 S10 peptide as a positive control (data not shown). Two different inhibitors to Aurora-A eradicated the H3 T118ph signal (*Figure 3—figure supplement 2A*). Upon Aurora-A knockdown, which eradicated most of the Aurora-A protein and activity (*Figure 3—figure supplement 2B*, *Figure 3—figure supplement 3A*), H3 T118ph was undetectable on chromatin (*Figure 3B*). In agreement with Aurora-A being the *bona fide* H3 T118 kinase, knockdown of TPX2, a known activator of Aurora-A (*Kufer et al., 2002*), greatly reduced H3 T118ph (*Figure 3—figure supplement 3B,C*). Taken together, these results demonstrate that Aurora-A mediates H3 T118 phosphorylation.

## H3 T118I and T118E increase lagging chromosomes and chromosome alignment errors

Given that H3 T118ph is detectable on chromatin during early mitosis (*Figure 1G*, *2B*), we investigated whether H3 T118ph plays a role in mitotic progression. To do this, we mutated T118 to alanine to prevent its phosphorylation. This serves as a negative control that is not expected to yield a phenotype, because there is still phosphorylation of the endogenous H3. We also mutated T118 to glutamic acid (E), although this mutation does not cause the nucleosome destabilization or altered nucleosome structures that result from T118 phosphorylation in vitro (*North et al., 2011*, *North et al., 2014*). As such, T118E is not an effective mimic of T118 phosphorylation, at least on mononucleosomes in vitro. We also mutated H3 T118 to isoleucine (I) to recapitulate the yeast sin mutant. Transient transfection of HEK 293TR cells with plasmids expressing histone H3:YFP where T118 was mutated to E or I led to a significantly increased incidence of lagging chromosomes (*Figure 3C,D*). Equal expression of the wild type and mutant H3 proteins was verified by western blot analysis (*Figure 3—figure supplement 4A*). Using time-lapse microscopy, we found that cells expressing H3 T118E:YFP or H3 T118I:YFP had significant delays in cytokinesis (*Figure 3E*, *Figure 3—figure supplement 4B*, see Materials and methods). Furthermore, whenever a lagging chromosome was evident, there also was an accompanying delay in the subsequent cytokinesis, regardless of the transfected construct (*Figure 3—figure supplement 4C*). From these experiments, we conclude that expression of H3 T118I and T118E results in lagging chromosomes that delay cytokinesis.

An increase in lagging chromosomes is symptomatic of defects in chromosome congression (*Thompson and Compton, 2011*). This prompted us to investigate if phosphorylation of H3 T118 plays a role in correction of chromosome alignment errors, using an error correction assay (*Lampson et al., 2004*, *Santaguida et al., 2010*). For this, we created a panel of stable 293TR cell lines expressing FLAG-tagged wild type H3, H3 T118E, T118I, or T118A from the same locus. All the

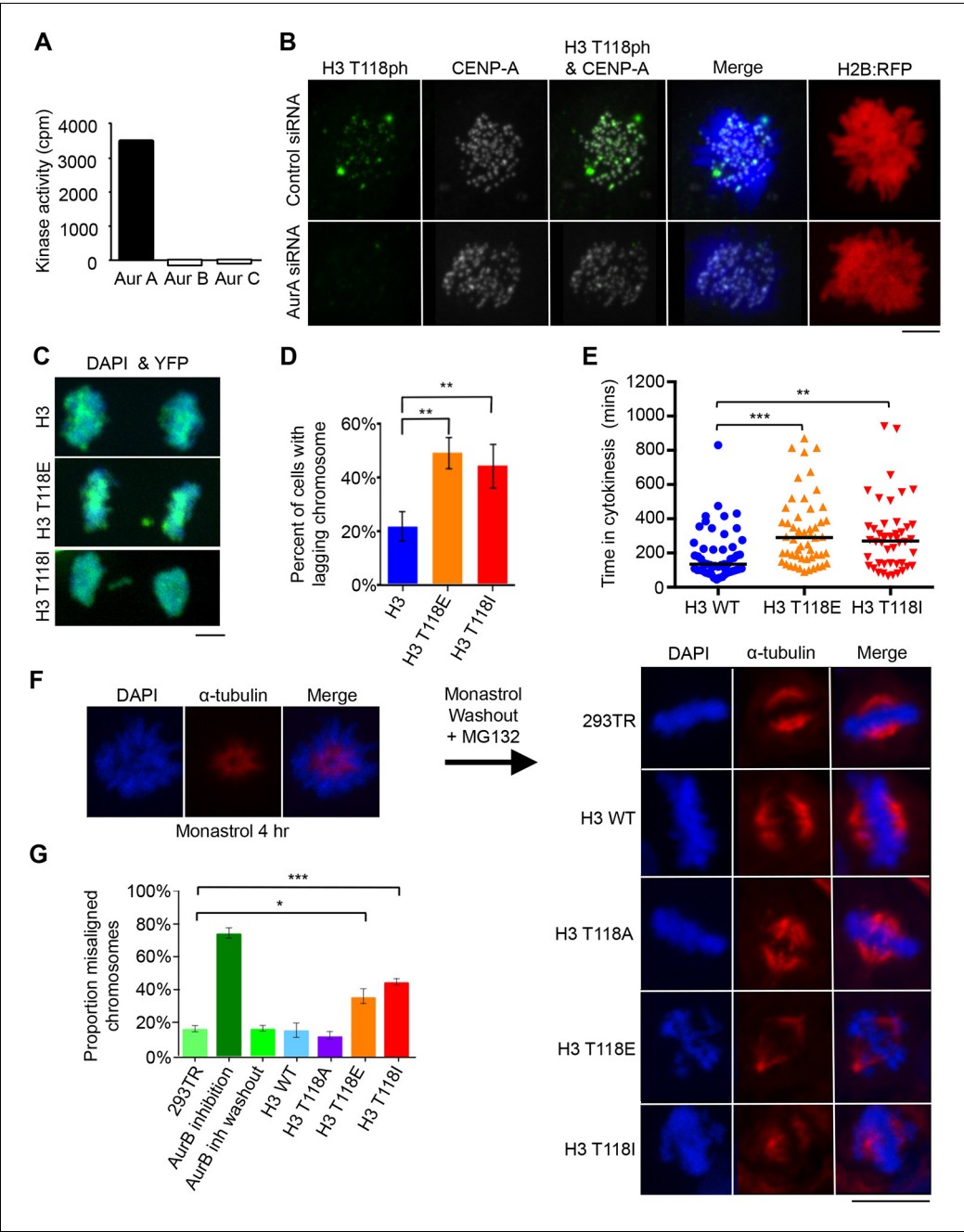

**Figure 3.** Aurora-A phosphorylates H3 T118 and mutations that mimic T118 phosphorylation cause mitotic defects. (**A**) *In vitro* kinase activity of Aurora-A,-B,-C for H3 T118 peptide. (**B**) Immunofluorescence of pro-metaphase HeLa cells cotransfected with H2B:RFP and siRNA to Aurora-A (bottom) or control scrambled siRNA (top). Scale bar = 5 μm. (**C**) Cytokinesis in 293TR cells transiently transfected with H3-YFP plasmids. YFP (yellow) and DNA stained with DAPI (blue). Scale bar = 5 μm. (**D**) Quantitation of C (n=30 cells in anaphase, **p=0.01, by Fishers exact test). Error bars represent SD of the mean (SDM). (**E**) Quantitative data of live cell imaging showing differences in average length in cytokinesis during live cell imaging (n = 50 cells, **p<0.01 and ***p<0.001 by unpaired student t-test). (**F**) Error correction assay for 293TR stable cell lines expressing H3. Inhibition of Aurora-B with ZM447439 represents an extreme case of inability to correct error. Scale bar =10 μm (**G**) Quantitation of cells with misaligned chromosomes on the metaphase plate as in F. (*p<0.05 and ***p<0.001 by Fishers exact test). Error bars represent SDM.

The following figure supplements are available for figure 3:

*Figure 3 continued on next page*

*Figure 3 continued*

**Figure supplement 1.** Results of in vitro kinase screen on peptide spanning H3 T118.

**Figure supplement 2.** Aurora-A inhibitors lead to decreased H3 T118ph.

**Figure supplement 3.** Knockdown of TPX2 leads to reduced H3 T118ph.

**Figure supplement 4.** Characterization of transient transfections and stable cell lines of wild type and mutant H3.

**Figure supplement 5.** FLAG-tagged wild type and mutant H3 are equally incorporated into chromatin.

**Figure supplement 6.** Overview of the *Drosophila* system expressing wild type and mutant H3 proteins.

H3:FLAG proteins were expressed to equivalent levels, at approximately 10% of the endogenous H3 level (*Figure 3—figure supplement 4D,E*) and all could be incorporated into chromatin (*Figure 3—figure supplement 5A*). We further verified that the H3 T118 mutations in H3:FLAG did not cause a delay in prophase to anaphase (*Figure 3—figure supplement 5B*). The error correction assay was as follows: Monastrol was used to induce a monopolar spindle and kinetochore-microtubule attachment errors (*Figure 3F*). The cell lines were able to recover by washing out Monastrol if proper check-points and machinery are in place and the chromosomes will attach to bipolar spindles. Additionally, cells were released in the presence of MG132 to allow time to align the chromosomes to the meta-phase plate by preventing cells from entering anaphase. Importantly, the H3 T118 mutations did not delay release from the pro-metaphase arrest (*Figure 3—figure supplement 5C*). Expression of either H3 T118E or T118I significantly decreased the ability to align chromosomes compared to wild type H3 or T118A (*Figure 3F,G*). This result suggests that an over abundance of H3 T118E and T118I mutants may hinder chromosome congression.

## Normal levels of H3 T118 phosphorylation are essential for development

Our results suggest an important role for phosphorylation of H3 T118 in regulating chromosomal dynamics in metazoans. However, these studies were performed in a situation where only 10% of the histone H3 was mutant. In order to examine the consequences of having all or none H3 phos-phorylated on T118, we introduced the T118 mutations into 12 copies of the H3 gene on transgenes and introduced them into *Drosophila* where the endogenous H3 gene copies were deleted (*Figure 3—figure supplement 6A*), such that the flies only expressed H3 T118A, T118E, or T118I (*Gunesdogan et al., 2010*). While control animals bearing wild type H3 survived to adulthood, animals expressing the mutant H3 T118A, E and I died as embryos after depletion of the maternal contribution of histones (*Figure 3—figure supplement 6B*). These results indicate that normal levels of H3 T118ph are essential for development.

## H3 T118ph remains at misaligned chromosomes

Having found that phosphorylation of H3 T118 was essential for development in fruit flies, we sought to gain a better understanding of its function. Since cells expressing H3 T118I and T118E showed reduced chromosome congression, we asked if H3 T118ph remains at centromeres of misaligned chromosomes as the cells enter metaphase. Caffeine was used to induce misaligned chromosomes (*Katsuki et al., 2008*). H3 T118ph remained at centromeres of misaligned chromosome along with the spindle assembly checkpoint (SAC) kinase BubRI, even in metaphase (*Figure 4A*). This suggests that removal of H3 T118ph is triggered by chromosome alignment and led us to speculate that H3 T118ph plays a role in achieving efficient chromosome attachment. Accordingly, we investigated the potential molecular reasons for the defect in chromosome congression caused by H3 T118E and T118I. Outer-kinetochore proteins, spindle assembly checkpoint proteins and the heterochromatin landscape were indistinguishable between cells expressing H3 T118A, T118E, T118I or wild type H3 (data not shown). Taken together, these data show that misaligned chromosomes in H3 T118I and T118E mutants are capable of forming proper kinetochores and recruiting SAC proteins

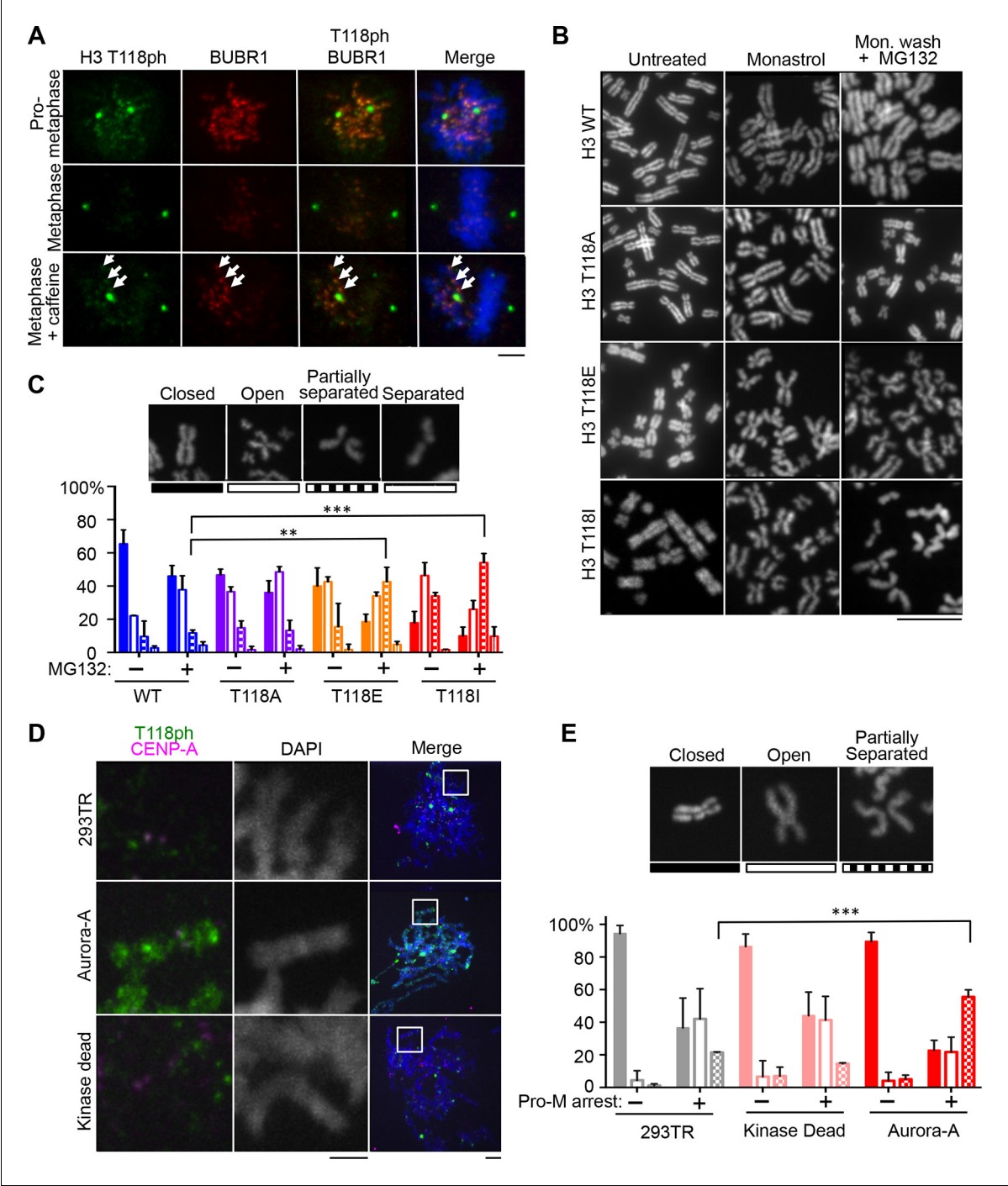

**Figure 4.** H3 T118I, T118E and Aurora-A overexpression lead to premature loss of cohesion. (**A**) Immunofluorescence of HeLa cells representing pro-metaphase (top panel), metaphase (middle panel) and caffeine-treated (bottom). The primary antibodies used were histone H3 T118ph (green), BubR1 (red) and DNA was stained with DAPI (blue). Scale bar = 5 µm. (**B**) Chromosome spreads of H3 T118 mutant cell lines following the error correction assay either untreated (left) or monastrol (middle) then released into MG132 (right). Scale bar = 5 µm. (**C**) The degree of cohesion loss for Monastrol (-) and Monastrol washout MG132 (+) treatments were scored from B. (n=100 cells per treatment collected over 3 experiments, \*\*p<0.01 and \*\*\*p<0.001 by unpaired student t-test). Error bars represent the SDM. (**D**) Chromosome spreads of 293TR cell lines with over expression of Aurora-A or Aurora-A KD. The primary antibodies used were against CENP-A (magenta), H3 T118ph (green), and DNA was stained with DAPI (blue). Scale bar = 5 µm. (**E**) Quantitation of Fig. S4C colcemid pro-metaphase arrest (Pro-M). (n=100 cells per treatment, collected over 3 experiments \*\*\*p<0.001 by unpaired student t-test). Error bars represent the SDM.

*Figure 4 continued on next page*

*Figure 4 continued*

The following figure supplements are available for figure 4:

**Figure supplement 1.** Metaphase spreads of 293TR stable cell lines expressing wild type H3 or mutant H3 proteins, to demonstrate cohesion defect upon prolonged pro-metaphase arrest.

**Figure supplement 2.** Characterization of 293TR stable cell lines expressing wild type Aurora-A:FLAG and Aurora-A Kinase Dead:FLAG.

## Loss of cohesion due to excess H3 T118ph

Because sister chromatid cohesion is important for chromosome congression, we examined whether the H3 T118I or T118E mutations caused faulty cohesion. Mitotic spreads from cells expressing wild type H3 or H3 T118A upon pro-metaphase arrest (induced by Monastrol) and metaphase arrest (Monastrol arrest released into MG132) mostly had closely associated sister chromatids with 'closed' or 'open' arms (*Figure 4B,C*, *Figure 4—figure supplement 1*). By contrast, H3 T118E or H3 T118I caused a higher incidence of chromosomes with "partially separated" arms, indicating loss of arm cohesion and partial loss of centromeric cohesion (*Figure 4B,C*). Similar defects in cohesion were observed for cells expressing T118E and T118I upon pro-metaphase arrest with the microtubule destabilizing drugs nocodazole and colcemid (data not shown). The loss of cohesion was most pronounced for H3 T118I, where partially separated sister chromatids were predominant in 50% of the cells versus 10% of the cells for wild type H3 (*Figure 4C*). Furthermore, the proportion of cells where most of the sister chromatids were totally separated, indicating complete loss of cohesion, was 16% for H3 T118I versus 4% for wild type H3 (*Figure 4C*). These data indicate that H3 T118E or H3 T118I promotes loss of cohesion at the centromere and chromosome arms. Given the correlation between faulty cohesion and chromosome alignment defects, we propose that the faulty cohesion caused by expression of H3 T118I or T118E is responsible for the defects in chromosome alignment.

Aurora-A overexpression has been linked to aneuploidy and cancer, presumably through its role in centrosome duplication. Aurora-A overexpression has not been linked to cohesion loss previously, but this could provide an alternate explanation for aneuploidy. Therefore, we made isogenic cell lines overexpressing Aurora-A and kinase dead Aurora-A. The cell lines had equal expression of Aurora-A (*Figure 4—figure supplement 2A*) and proceeded relatively normally through the cell cycle (data not shown). Overexpression of Aurora-A increased levels of H3 T118ph along the chromosome arms (*Figure 4D*). We asked if overexpression of Aurora-A recapitulates the loss of cohesion caused by expression of H3 T118I and T118E. Upon colcemid-induced pro-metaphase arrest, overexpression of Aurora-A caused 44% of the sister chromatids to be "partially separated" as compared to 20% for the control cell line (*Figure 4E*, *Figure 4—figure supplement 2B*). Because overexpression of Aurora-A leads to cohesion loss, it is likely that cohesion loss in the H3 T118E and T118I mutants is due to their structurally mimicking elevated levels of H3 T118ph.

## Excess H3 T118ph leads to defective chromosomal condensation

Because cohesion defects can be caused by altered chromatin integrity, we measured the length and width of chromosome one from each H3 mutant. We identified chromosome one by using a special DAPI-treatment protocol to highlight the large pericentromeric heterochromatin cluster (*Figure 5—figure supplement 1A*). Expression of H3 T118E and T118I made chromosome one significantly wider and shorter (*Figure 5A*, *Figure 5—figure supplement 1B,C*). To investigate centromere integrity in the T118 mutants, we measured the sister chromatid interkinetochore distance in chromosome spreads collected after arrest in metaphase. We found that H3 T118E and T118I significantly increased sister chromatid interkinetochore distances (*Figure 5B*), as measured by immunostaining for CENP-A (*Figure 5—figure supplement 1D*).

To obtain a higher resolution view of the effects of the H3 T118 mutations on chromosome structure, we performed scanning electron microscopy (SEM). Upon pro-metaphase arrest, chromosomes from the H3 wild type and T118A mutant cell lines were organized into loops and coils to form very tight compact structures (*Figure 5C*). By contrast, mitotic chromosomes from the H3 T118E and T118I cell lines were less tightly packed with longer radiating DNA loops. These results indicate that H3 T118E and T118I disrupt the higher order chromatin packaging. This grossly altered mitotic

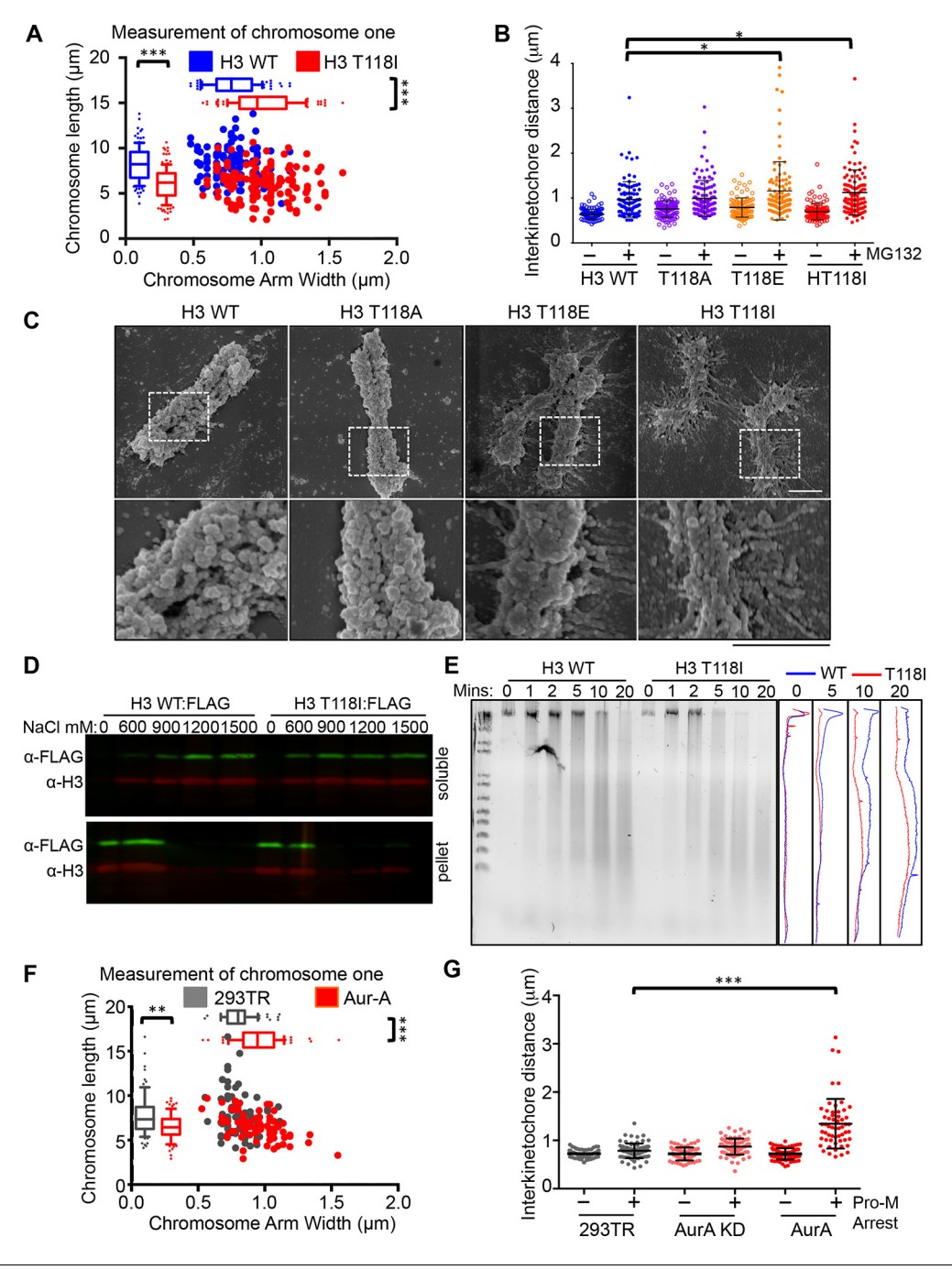

**Figure 5.** Altered chromosomal compaction due to H3 T118E, H3 T118I or overexpressing Aurora-A. (**A**) Measurement of the width and length of chromosome one for over 50 chromatids for each H3 WT:FLAG and H3 T118I:FLAG stable cell lines (***p<0.001 by Wilcoxon rank sum test). (**B**) Interkinetochore distances for pairs of sister chromatids. N=100 centromeres from 5 mitotic chromosome spreads (*p<0.01 by student t-test). Error bars represent SD of the mean (SDM). (**C**) SEM images taken at 50 K and 100 K magnification upon prolonged mitotic arrest. Scale bar = 1 μm. (**D**) Western analysis of soluble (free histones) and pellet (chromatin) fractions following successive increasing concentration NaCl extractions. (**E**) Dnase-I digestion analysis on nocodazole arrested cells. Densitometric profiles are shown on the right. (**F**) As in A, comparing 293TR versus Aurora–A overexpressing cell lines for over 30 chromatids (***p<0.001 and **p<0.01 by Wilcoxon rank sum test). (**G**) As in B, comparing 293TR cell lines with over expression of Aurora-A or Aurora-A KD with and without colcemid arrest (Pro-M arrest, pro-metaphase arrest). N=50 centromeres from 5 mitotic spreads (***p<0.001 by student t-test). Error bars represent SDM.

*Figure 5 continued on next page*

*Figure 5 continued*

The following figure supplements are available for figure 5:

**Figure supplement 1.** The interkinetochore distance becomes longer upon expression of T118I.

**Figure supplement 2.** The Aurora-A kinase dead does not change the packaging of chromosome 1, as compared to expression of Aurora-A.

chromosome structure led us to test whether the H3 T118I mutation causes the histones to be more readily removed from chromatin. In agreement, H3 T118I was more readily extracted from chromatin than wild type H3 at 600 mM salt (*Figure 5D*). Expression of H3 T118I also increased DNA accessibility to the nuclease DNase I in both asynchronous and mitotically arrested cells (*Figure 5E*, *Figure 5—figure supplement 2A*). Together, these results are consistent with biochemical studies that showed that mononucleosomes with H3 T118ph favor the removal of histone H3 from DNA compared to unphosphorylated mononucleosomes (*North et al., 2011*).

Given that overexpression of Aurora-A results in excess H3 T118ph (*Figure 4D*), we asked if it also disrupts chromosome integrity. Overexpression of Aurora-A caused significant widening and shortening of the chromosome arms of metaphase chromosomes (*Figure 5F*) as was observed for H3 T118E and T118I (*Figure 5A*), while overexpression of Aurora-A KD did not (*Figure 5—figure supplement 2B*). Overexpression of Aurora-A also caused increased sister chromatid interkinetochore distances (*Figure 5G*). These results further indicate that the H3 T118I and T118E mutations are functional mimics of H3 T118 phosphorylation in vivo, and show that H3 T118ph disrupts higher order chromatin packaging.

## H3 T118I and T118E cause premature removal of cohesin from DNA

The altered chromatin integrity and cohesion defect caused by excess H3 T118ph or mutations that mimic excess H3 T118ph led us to ask whether there was a dissociation of cohesin proteins from DNA due to excess H3 T118ph. During mitotic delay, the intensity of the Rad21/Scc1 component of the cohesion complex along chromosome arms and at centromeres was drastically reduced in cells expressing H3 T118E and T118I (*Figure 6A*, *Figure 6—figure supplement 1*). Mechanistically, the loss of cohesin and the resulting faulty cohesion phenotype that is caused by excess H3 T118ph (*Figure 4B,C,E*) could result from multiple causes: premature activation of separase, premature removal of cohesion via cohesin phosphorylation, or improper establishment of cohesion. We set out to distinguish amongst these possibilities. To ask if cells expressing H3 T118I and T118E had premature activation of separase during mitotic delay, we analyzed mitotic spreads after incubation with MG132, which prevents degradation of Cyclin B and Securin and therefore inhibits separase activation (*Rock et al., 1994*). The fact that the T118I and T118E mutants still displayed cohesion loss, despite inhibition of separase (*Figure 6B*), indicates that cohesin loss in the T118E/I mutants is not due to premature separase activity. The bulk of cohesion is removed in pro-metaphase by phosphorylation of the cohesin subunit SA2 by PLK-1 kinase or Aurora-B kinase (*Hauf et al., 2005*). We found that the PLK-1 inhibitor, BI2536, and the Aurora-B inhibitor, hesperidin, prevented cohesion loss in all the H3 expressing cell lines (*Figure 6B*). Sister chromatid cohesion is also facilitated by DNA catenation during DNA replication (*Nitiss, 2009*). To prevent DNA decatenation, we used a specific inhibitor of Topo II, ICRF-193 and found that chromosomes became extremely tangled, indicative that DNA catenation is undisturbed by the H3 T118 mutations (*Figure 6B*). Taken, together, these data indicate the H3 T118I and T118E mutations do not disrupt the proper establishment of sister chromatid cohesion by both DNA and sister chromatid catenation, but are likely to lead to premature cohesion loss via the PLK-1 or Aurora-B mediated pathway.

## Premature loss of condensin I from DNA due to H3 T118E, T118I and overexpression of Aurora-A

During our PLK-1 inhibition studies, we observed that chromosomes from cells expressing H3 T118I were extremely short (*Figure 7A*, *Figure 7—figure supplement 1A*), a phenotype observed previously (*van Vugt et al., 2004*). These short chromosomes occurred in 90% of the mitotic spreads

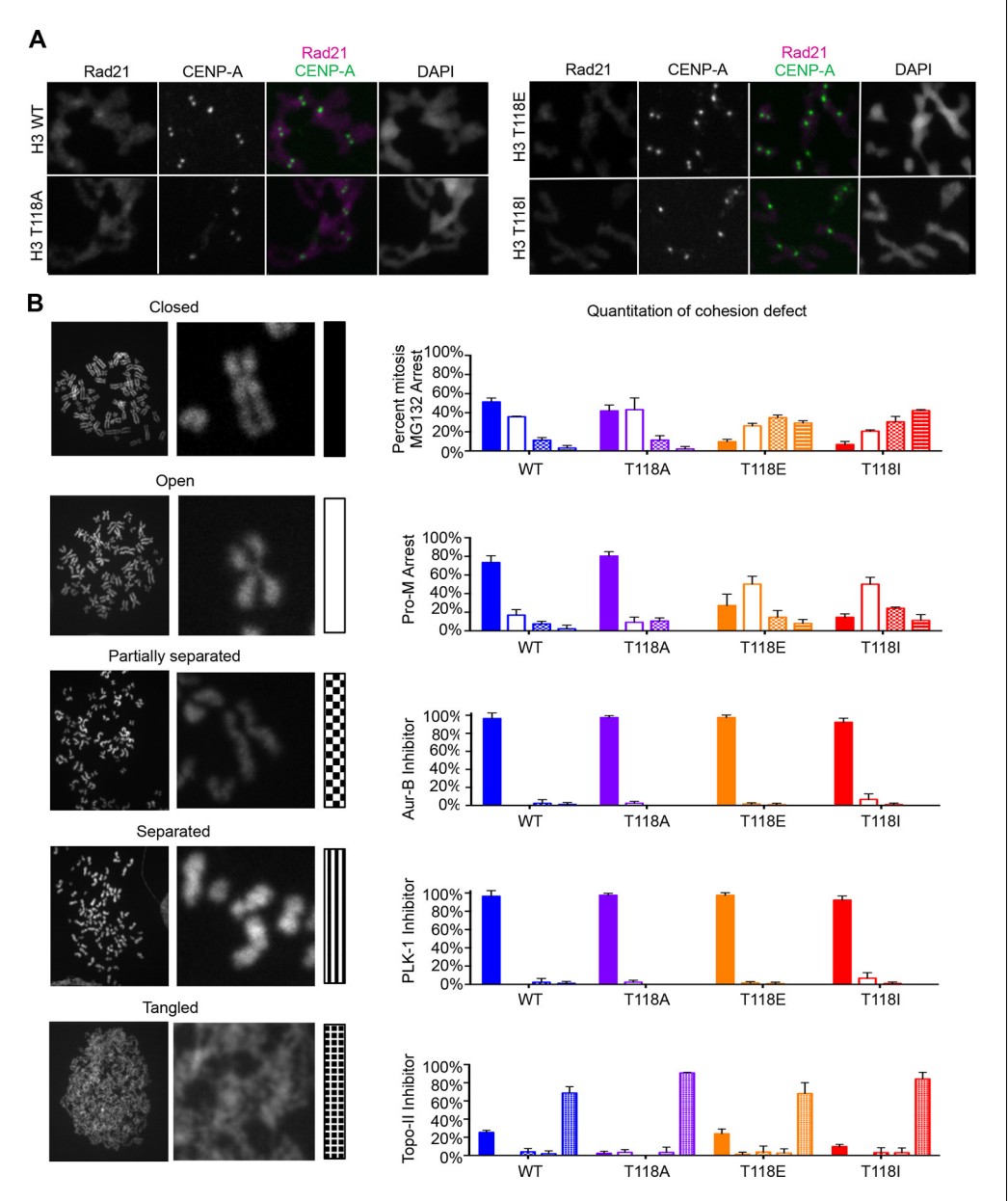

**Figure 6.** Premature cohesion loss in the phosphomimetic and SIN mutants is independent of separase activity, but dependent on proper centromere tension. (**A**) Mitotic spreads following the error correction assay. The primary antibodies used were against Rad21, cohesion subunit (magenta), CENP-A (green), and DNA was stained with DAPI (blue). Scale bar = 5 μm. (**B**) Quantitation of the degree of cohesion loss for H3:FLAG stable cell lines, upon proteasome inhibition with MG132, treatment with colcemid, Aurora-B (hesperidin), Plk-1 (BI-2536), and Topo-II (ICRF-193) inhibitors for 3 hr was scored (n=75 cells, per treatment collected over 3 experiments). Insets show representative chromosomes for each type of defect: closed, open, partially separated, separated or tangled.

The following figure supplement is available for figure 6:

**Figure supplement 1.** Stable cell lines expressing H3 T118 mutants do not alter Rad21 staining in an asynchronous cell population.

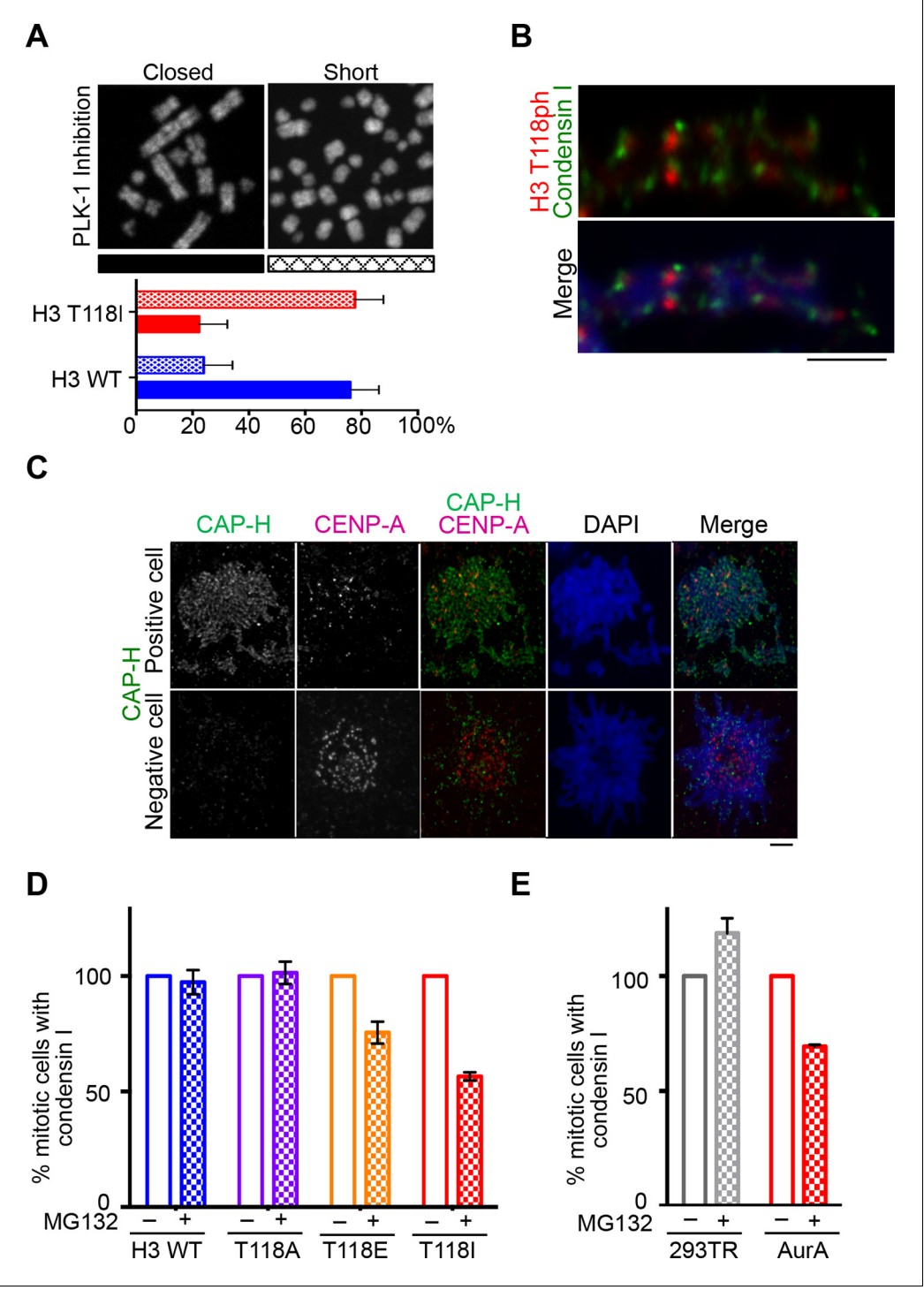

**Figure 7.** Reduced condensin I association with chromatin due to H3 T118E and T118I. (**A**) Chromosome spreads upon PLK-1 inhibition and quantitation of the degree of cohesion loss for H3: WT:FLAG and H3 T118I:FLAG stable cell line. Insets show representative chromosomes for each type of defect: closed and short. (n=50 cells). Scale bar = 5 µm. (**B**) Extended chromatin fibers from 293TR CAP-H:tGFP cells. Scale bar = 2 µm. The primary antibodies used were against tGFP (green), H3 T118ph (red), and DNA was stained with DAPI (blue). (**C**) Representative mitotic spreads for condensin I (CAP-H:tGFP) positive and tGFP negative cell. The primary antibodies used were against tGFP (green), CENP-A (red), and DNA was stained with DAPI (blue). Scale bar = 5 µm. (**D**) Quantitation of number of cells with positive condensin I (CAP-H:tGFP) for mutant H3 stable cell lines treatment without Monastrol (-) and Monastrol washout followed by MG132 (+) treatments. SDM is for three independent experiments (n=100

*Figure 7 continued on next page*

*Figure 7 continued*

per treatment). (E) As in D, quantitation using 293TR and Aurora-A overexpressing cell line from over 50 mitotic spreads in each condition. Error bars are SDM.

The following figure supplements are available for figure 7:

**Figure supplement 1.** Topoisomerase II and H3 T118ph display different localization patterns along chromatin fibers.

**Figure supplement 2.** Topoisomerase II and its levels are unaltered in chromatin from cell lines expressiong H3 WT:FLAG, H3 T118A:FLAG, H3 T118E:FLAG and H3 T118:FLAG.

**Figure supplement 3.** Condensin II and its levels are unaltered on chromatin from cell lines expressing H3 WT:FLAG, H3 T118A:FLAG, H3 T118E:FLAG and H3 T118I:FLAG

**Figure supplement 4.** The binding of Condensin I to nucleosomes is not affected by H3 T118 mutations.

---

from PLK-1 inhibited cells expressing H3 T118I compared to 22% for wild type H3. This hypercondensation phenotype suggests that H3 T118I may disrupt chromosome scaffolding proteins involved in shaping mitotic chromosomes, including condensin I and II and Topo II. However, H3 T118ph does not co-localize with Topo II (*Figure 7—figure supplement 1B*) or condensin I (*Figure 7B*). Next, we determined whether the amounts of the scaffold proteins condensin I, condensin II and Topo II were altered on mitotic chromosomes in the H3 T118 mutants. The staining of Topo II (*Figure 7—figure supplement 2A,B*) and condensin II (*Figure 7—figure supplement 3A,B*) was similar among cells expressing wild type or mutant H3. However upon mitotic delay, by the error correction assay, there was a significant loss of turbo-GFP (tGFP) tagged condensin I CAP-H protein in both H3 T118E (25% of mitotic cells were tGFP negative) and T118I (50% of mitotic cells were tGFP negative) cell lines as compared to wild type H3 (0% of mitotic cells were GFP negative) (*Figure 7C,D*). These data demonstrate that H3 T118I and T118E results in reduced levels of condensin I, but not condensin II or Topo II, on chromatin, suggesting that H3 T118ph plays a role in reducing condensin I occupancy on the chromatin.

Given that mutations that mimic H3 T118ph had reduced condensin I occupancy, we asked whether H3 T118ph directly prevents the binding of condensin I to chromatin. We purified the condensin I complex (*Figure 7—figure supplement 4A*) and used expressed protein ligation to generate mononucleosomes that were 100% phosphorylated on H3 T118 (*North et al., 2011*). The histones carrying H3 T118ph generated not only canonical nucleosomes, but also altosomes and disomes (*Figure 7—figure supplement 4B*) as seen previously (*North et al., 2014*). In electrophoretic mobility shift assay (EMSA) at higher levels of condensin 1, we found that condensin I could bind to nucleosomes and the altered histone-DNA forms, irrespective of the phosphorylation status of H3 T118. This result indicates that H3 T118ph does not directly affect condensin I binding to a mononucleosome. As such, we favor the idea that H3 T118ph promotes changes in global chromatin packaging that may indirectly reduce condensin I occupancy. Therefore, we asked if overexpression of Aurora-A recapitulates the loss of condensin I caused by expression of H3 T118I and T118E. Indeed, upon mitotic delay, there was a significant loss of condensin I from chromatin upon Aurora-A overexpression (25% of mitotic cells were GFP negative) as compared to the control (0% of mitotic cells were GFP negative) (*Figure 7E*). This result shows that excess H3 T118ph leads to condensin I loss from chromatin. Taken together, these data suggest that the function of mitotic H3 T118ph is to indirectly reduce condensin I and cohesin occupancy on chromatin via its influence on chromosome packaging.

## Discussion

Here we provide the first in vivo characterization of phosphorylation on threonine 118 of histone H3 (H3 T118ph), a modification that breaks histone-DNA contacts at the nucleosomal dyad. In metazoans, H3 T118ph is dynamically regulated through mitosis by the Aurora-A kinase, occurring at pericentromeric regions and at discrete locations on chromosome arms. Excess H3 T118ph (achieved by

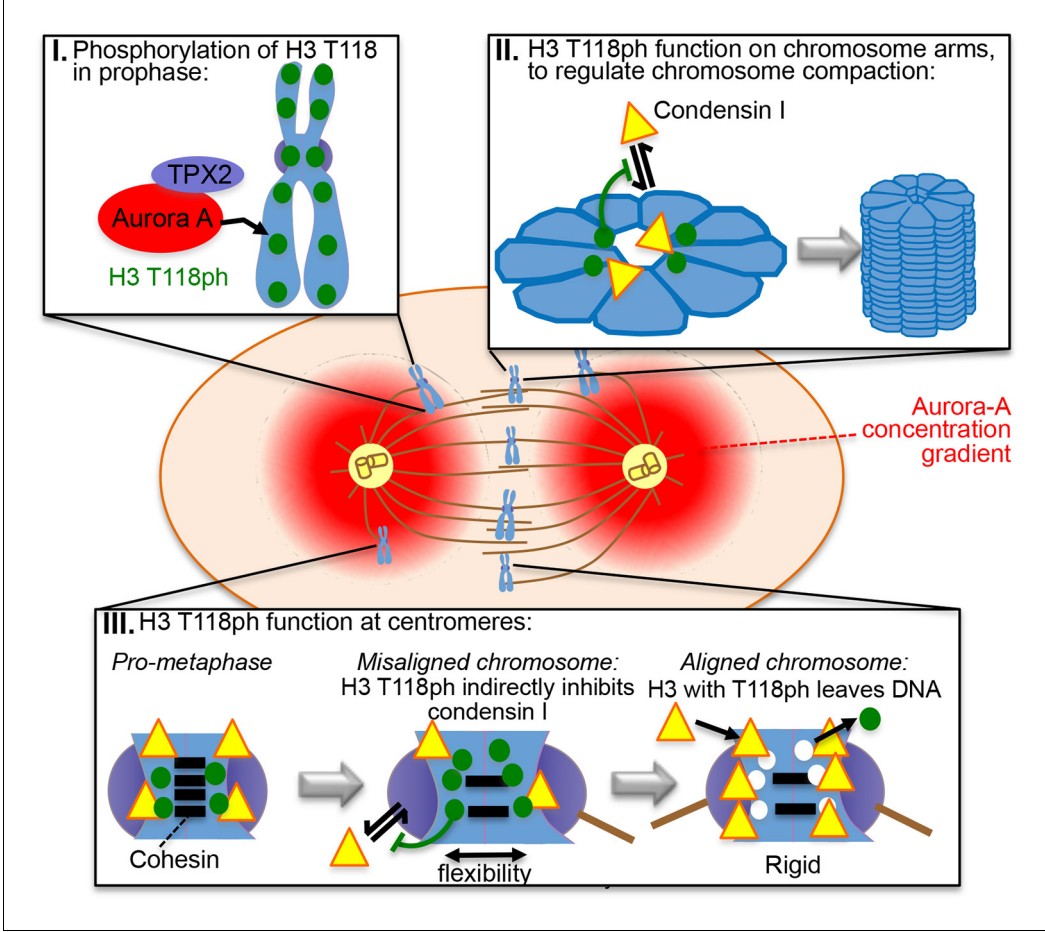

**Figure 8.** Model for the functions of H3 T118ph as explained in the text.

overexpression of Aurora-A or mimicked by amino acid substitution) resulted in increased numbers of lagging chromosomes, defects in chromosome congression, delayed cytokinesis, altered chromosome compaction, cohesion loss and cohesin and condensin I loss. Normally, H3 T118ph disappears from each chromosome when chromosome alignment is achieved. Given that condensin I increases the rigidity at the centromere (*Ribeiro et al., 2009*), we propose a model where H3 T118ph alters the chromatin structure to limit condensin I and cohesin occupancy in order to enable efficient attachment to the mitotic spindle and effective chromosome compaction (*Figure 8*).

Aurora-A is best known for its role in centrosome separation (*Dutertre et al., 2002*). Our data further supports Aurora-A's involvement in chromosome error correction (*Chmátal et al., 2015*; *Ye et al., 2015*), specifically through its role in phosphorylating H3 T118 on chromatin. In agreement, the H3 protein sequence K-R-V-T-I fits the R/K/N-R-X-S/T-B consensus site for Aurora-A (where B is a hydrophobic residue) (*Ferrari et al., 2005*). We propose that Aurora-A, in partnership with its activator TPX2, is responsible for phosphorylation of H3 T118 on the chromatin arms and centromeres. Fittingly, Aurora-A is detectable at the centromere during mitosis (Chmatal, Yang et al. 2015). Precedent exists for Aurora-A-mediated phosphorylation of centromeric proteins, on substrates including NDC80 (Ye, Deretic et al. 2015), CENP-A (*Kunitoku et al., 2003*) and CENP-E (*Kim et al., 2010a*). In addition to being found at the centromere, H3 T118ph also occurred in a periodic punctuate pattern along the chromosome arms. As such, H3 T118 is the first known target for phosphorylation by Aurora-A along chromosome arms. This raises the question of how Aurora-A/TPX2 is directed to phosphorylate H3 T118 along the chromosome arms. In order to gain insight into this mechanism, we attempted to map exactly where H3 T118ph occurs on chromatin by ChIP-seq, but unfortunately the antisera against H3 T118ph failed in ChIP analysis (data not shown). This

was disappointing since we were able to successfully immunoprecipitate H3 T118ph with the antibody (*Figure 1F*), raising the possibility that the antibody immunoprecipitates only H3 T118ph on free histones. Regardless of how Aurora-A is targeted to chromatin, it is likely that the phosphorylation of H3 T118ph will also require nucleosome disruption by an ATP-dependent nucleosome remodeler given the buried location of this residue within the nucleosome structure. As such, this would provide an additional step to tightly regulate the function of this key histone post-translational modification.

In addition to the insight gained from the location and timing of H3 T118ph, much of our understanding of H3 T118ph function comes from the analysis of histone mutants. Given that we can only express the H3 mutants to be approximately 10% of the total histone H3 in human cells, it is not surprising that the H3 T118A loss of function mutant gave no detectable phenotype in the presence of endogenous H3 T118ph. In contrast, the H3 T118E and T118I mutants clearly had dominant effects on wild type histones in human cells. It should be noted that *Drosophila* cell clones expressing only H3 T118A, H3 T118E, or H3 T118I have significant defects in cell growth (Graves et al., submitted). H3 T118ph acts to physically distort the nucleosomal DNA at the nucleosome dyad in order to loosen the nucleosome structure and generate altered nucleosomal states (*North et al., 2014*). Our results show that the alterations to the nucleosome structure that are induced by H3 T118ph impact higher order levels of chromosome packaging. This structural role of phosphorylation of H3 T118 can explain why the T118I mutant gave even more drastic phenotypes than T118E in our experiments. Although isoleucine is not the classic phosphomimetic substitution, it does have a large bulky side chain that would distort the trajectory of the nucleosomal DNA around the histone octamer to an even greater extent than the traditional phosphomimetic of glutamic acid. Furthermore, isoleucine mimics the rigidity of phosphate group compared to the flexible side chain of glutamic acid. Functional support for the idea that H3 T118I structurally mimics the effect of phosphorylation of T118 comes from the fact that overexpression of the T118 kinase, Aurora-A, leads to identical phenotypes to T118I.

## H3 T118ph function at the centromere

H3 T118ph appears at pericentromeric regions during prophase and disappears from each chromosome as it aligns at the metaphase plate. Furthermore, the H3 T118I and T118E mutants resulted in displacement of condensin I and cohesin from chromatin and generated chromosomes with looser chromatin packaging. Accordingly, we propose that H3 T118ph plays an important role in organizing the chromatin structure around centromeres to achieve optimal levels of cohesin and condensin I association to permit enough conformational flexibility for microtubule attachment. Condensin I is highly enriched at centromeres in mitosis in metazoans (*Kim et al., 2013*) and promotes the rigidity of the centromere (*Gerlich et al., 2006*). Additionally, Aurora–A has been demonstrated to play a role in error correction by destabilizing microtubule connections of misaligned chromosomes. Upon knockdown or inhibition of Aurora-A the kinetochore, as well as their attachment to microtubules, become more rigid and stable (Chmatal, Yang et al. 2015, Ye, Deretic et al. 2015). As such, H3 T118ph at the centromere appears to act to limit condensin I occupancy in order to increase flexibility at the centromeres of misaligned chromosomes (*Figure 8*). This idea is supported by chromosomes from cells expressing H3 T118E, T118I or overexpressing Aurora-A having increased interkinetochore distances (*Figure 5B,G*), which could be indirectly or directly related to the role of H3 T118ph in removal of cohesin and condensin I. However, upon attachment to mitotic spindles from opposite centrosomes, the centromeric regions have to be rigid enough to resist the forces that the microtubules exert on the centromere in order to prevent separation of sister chromatids until anaphase (*Musacchio and Salmon, 2007*). Removal of H3 T118ph as soon as tension is sensed across the kinetochores would allow for better centromere rigidity. Consistent with an important role for H3 T118ph in achieving appropriate microtubule attachment, H3 T118ph remained at centromeres of misaligned chromosomes (*Figure 4A*).

## H3 T118ph function on chromosome arms

H3 T118ph occurs in a punctate periodic pattern along chromosome arms in prophase and prometaphase. Excess H3 T118ph (due to overexpression of Aurora-A or mutations that mimic the effect of phosphorylation) leads to gross alterations in chromosome compaction, with wider and

shorter chromosome arms and longer, less organized chromatin loops (*Figure 5*), suggesting that H3 T118ph plays a role in shaping mitotic chromosomes. Mitotic chromosomes have been suggested to be packaged in a two phase process (*Naumova et al., 2013*). In the first phase, a linear array of chromatin loops form at random, but consistent, positions along the chromosome. In the second phase, the loops longitudinally condense around the axes. These two different phases are mediated by the condensins, where condensin II is required for linear compaction along the chromosome axes while condensin I helps organize chromatin loops around the axes (*Shintomi and Hirano, 2010*, *Green et al., 2012*). Although the timing of appearance of H3 T118ph and condensin I on chromosome arms is similar, their spatial localization along the arms are distinct (*Figure 7B*). As such, there is no evidence that H3 T118ph physically recruits or displaces condensin I from chromatin. Indeed, other proteins promote condensin recruitment in yeast including kinetochore proteins (*Tada et al., 2011*) and the Ku heterodimer complex, which functions in non-homologous end joining (*Tanaka et al., 2012*). Perhaps related to its recruitment mechanism, yeast condensin interacts with histone H2A and H2AZ in vitro (*Tada et al., 2011*) and cross-linking mass spectrometry studies have found interactions between condensin I and H2A and H4 (*Barysz et al., 2015*). In addition, our evidence indicates that H3 T118 phosphorylation is likely to regulate condensin I occupancy on the chromatin, given that expression of H3 T118I, T118E and overexpression of Aurora-A cause loss of condensin I from chromosome arms (*Figure 7C–E*). This disruption of condensin I function is in agreement with the longer loops of chromatin that were observed by SEM in the T118E and T118I mutants (*Figure 5C*). Consistent with the delayed cytokinesis that occurs upon condensin I knockdown (*Gerlich et al., 2006*) the H3 T118I and T118E mutants caused a delay in cytokinesis (*Figure 3E*). Given that condensin I interacts with chromosomes in a more dynamic manner than condensin II (*Gerlich et al., 2006*),we propose that the dynamic nature of the association of condensin I with chromatin enables H3 T118ph to regulate the levels of condensin I to shape the mitotic chromosomes as they condense. The ratio of condensin I to condensin II is very tightly controlled within cells, given that changes in the ratio profoundly alters the shape of mitotic chromosomes (*Shintomi and Hirano, 2010*, *Bakhrebah et al., 2015*). As such, the removal of H3 T118ph from the chromosome arms by metaphase, either by dephosphorylation or by our preferred model of physical removal of T118 phosphorylated H3 from the DNA, is likely to regulate the ratio of condensin I:condensin II for appropriate chromosome compaction. This function is likely to occur in an indirect manner via H3 T118ph affecting chromatin structure, given that condensin I binding to mononucleosomes is not affected by H3 T118ph in vitro (*Figure 7—figure supplement 4B*). Similarly, we propose that the loss of cohesin is an indirect consequence of the altered packaging of the chromatin structure caused by excess H3 T118ph, which may expose the cohesin ring to PLK-1 mediated phosphorylation and subsequent removal of cohesin (*Hauf et al., 2005*). However, we were unable to rule out the possibility that the cohesion phenotype may be due to loss of Sgo-1-mediated protection against PLK-1 and Aurora-B kinases.

Taken together, our work suggests a model where phosphorylation of H3 T118 at the nucleosome dyad by Aurora-A is a critical step to ensure chromosome congression, via its influence on chromosome compaction and cohesion through physically regulating nucleosome structure. These functions are likely to be conserved in metazoans, as we find similar localization and timing of H3 T118ph in nematodes, flies, and human cells. The importance of the ability to utilize H3 T118ph to alter the nucleosome structure to regulate mitosis is underscored by the embryonic lethality of flies where all of their histones are mutated to prevent T118 phosphorylation or to mimic persistent H3 T118 phosphorylation. Given that Aurora-A is overexpressed in many cancers, it is tempting to speculate that the carcinogenic effect of overexpressed Aurora-A may be mediated at least in part via altering the mitotic chromatin structure by phosphorylation at the nucleosome dyad.

## Materials and methods

### Constructs and cloning

Plasmid expressing human H2B:RFP was a kind gift from Walter Hittelman (MDACC, Houston, TX). Plasmids expressing human Aurora-A:FLAG and Aurora-A KD:FLAG were a kind gift from Subrata Sen (MDACC, Houston, TX) (*Katayama et al., 2012*). The CMV-histone *Drosophila* H3-YFP (dH3) plasmid was purchased from Addgene (plasmid 8694). The CapH:GFP plasmid was purchased from

Origene (Rockville, MD USA, RG201421). The shRNA histone H3 resistant plasmid pOZ-FH-C H3.1c: FLAG:HA (HuH3.1:FLAG) was kindly provided by Zhenkun Lou, Ph.D (Mayo Clinic, Rochester, Mn). Site directed mutagenesis was performed on the CMV-histone dH3-YFP and pOZ-FH-C H3.1c:FLAG: HA plasmids listed below using the QuickChange Site-directed Mutagenesis Kit (Agilent Technologies, Santa Clara, CA, USA 200515). This plasmid has the histone sequence of *Drosophila* histone H3 and corresponds to the human histone H3.2 amino acid sequence. The CMV-histone dH3 YFP T118A plasmid was generated using the following primers:

Forward: 5'- TTCATGCCAAGCGTGTCGCCATAATGCCCAAAGAC -3'
Reverse: 5'- GTCTTTGGGCATTATGGCGACACGCTTGGCATGAA -3'
The CMV-histone dH3 YFP T118E plasmid was generated using the following primers:
Forward:
5'- GCCATTCATGCCAAGCGTGTCGAGATAATGCCCAAAGACATCCAG -3'
Reverse:
5'- CTGGATGTCTTTGGGCATTATCTCGACACGCTTGGCATGAATGGC -3'
The CMV-histone dH3 YFP T118I plasmid was generated using the following primers:
Forward: 5'- TCATGCCAAGCGTGTCATCATAATGCCCAAAGACA -3'
Reverse: 5'-TGTCTTTGGGCATTATGATGACACGCTTGGCATGA -3'
The pOZ-FH-C HuH3.1T118A:FLAG primer was generated using the following primers:
Forward: 5'- CACGCTAAACGCGTCGCCATCATGCCCAAAG -3'
Reverse: 5'- CTTTGGGCATGATGGCGACGCGTTTAGCGTG -3'
The pOZ-FH-C HuH3.1T118E:FLAG plasmid was generated using the following primers:
Forward:
5'- GCTATTCACGCTAAACGCGTCGAGATCATGCCCAAAGATATCCAG -3'
Reverse:
5'- CTGGATATCTTTGGGCATGATCTCGACGCGTTTAGCGTGAATAGC -3'
The pOZ-FH-C HuH3.1T118:FLAG plasmid was generated using the following primers:
Forward: 5'- TCACGCTAAACGCGTCATCATCATGCCCAAAGATA -3'
Reverse: 5'- TATCTTTGGGCATGATGATGACGCGTTTAGCGTGA -3'
The following primers were used for a PCR ligation reaction to amplify HuH3.1:FLAG
Forward: 5'-ATGGCTCGTACGAAGCAAAC-3'
Reverse: 5'-CTAGGCGTAGTCGGGCACGTCGT -3'
The resulting PCR fragment was cloned into pcDNA5 FRT/TO TOPO TA plasmid (Life technology Grand Island, NY USA K6510-20)

## Antibodies and peptides

Mad2 antibody was a kind gift from Ted Salmon (UNC, Chapel Hill, NC). The following primary antibodies were purchased: polyclonal H3 T118ph (Abcam Cambridge, MA USA ab33310, lot 7 for western blots and lot 9 for immunofluorescence), H3S10ph (Abcam, ab14955), C-terminal H3 (Abcam, ab1791), N-terminal H3 (Active Motif Carlsbad, CA USA, 39763), γ-tubulin (Abcam, ab27074), CENP-A (Abcam, ab8245), CENP-A (Cell Signalling Technology Danvers, MA USA, 2186), GAPDH (Abcam, ab8245), M2-FLAG (Sigma St. Louis, MO USA, F3165), BubR1 (Abcam, ab4637), Hec1 (Abcam, ab3613), CENP-E (Abcam, ab4163), Hp1α (Active Motif, 39295), HP1β (Active Motif, 39979) HP1γ (Active Motif, 39981), Aurora-B/AIM-1 (BD Biosciences, 611082), SA2 (Bethyl laboratories, Montgomery, TX USA, A310-043A), Rad21 (Millipore Billerica MA, USA 05–908), H3 K9 me3 (Abcam, ab6001), CapD3 (Bethyl laboratories, A300-604A), Topo II (Milllipore, MAB4197), (phospho) Aurora-A T288 (Cell Signaling, 3079), Aurora-A Clone 35C1 (Invitrogen, 45–8900), α-tubulin (Sigma-Aldrich, T9026), α-tubulin (AbD Sterotec Raleigh, NC, USA MCA78G), anti-GFP (Roche Indianapolis, IN USA 11814460001), and anti-turboGFP (Origene TA150041).

The secondary antibodies used were as follows: Alexa Fluor 488 goat anti-rabbit (Life Technologies Carlsbad, CA, A11034), Alexa Fluor 594 goat anti-rabbit (Life Technologies, A11037), Alexa Fluor 488 goat anti-mouse (Life Technologies, A11029), Alexa Fluor 555 goat anti-rat (Cell Signaling), HRP-conjugated anti-mouse (Promega, Madison, WI USA, PR-W4011), and HRP-conjugated anti-rabbit (Promega, PRW4021).

Non-biotinylated peptides used were H3 unmodified (Abcam, ab12149), H3 S10ph (Abcam, ab1147), H3 K122ac (Abcam, ab34466), and H3 T118ph (Abcam, ab33505). Biotinylated peptides were either purchased from Anaspec (Freemont, CA, USA) or were a kind gift from Min Gyu Lee (MDACC).

## Cell lines and stable cell lines

HeLa cells were maintained in Dulbecco's Modified Eagle Medium (DMEM) supplemented with 10% fetal bovine serum and 1% penicillin/streptomycin. WI-38 cells were maintained in Eagle's Minimum Essential Medium (MEM) supplemented with 10% fetal bovine serum and 1% penicillin/streptomycin. MCF10A cells were maintained in DMEM/Nutrient Mixture F-12 supplemented with 5% horse serum, 1% penicillin/streptomycin, 10 mg/ml insulin, 1 mg/ml hydrocortisone, 25 µg/ml EGF, and 1 mg/ml cholera toxin. The Flp-in T-Rex 293 (293TR) cell line was purchased from Life Technologies (R780-07) and were maintained in DMEM 10% fetal bovine serum and 1% penicillin/streptomycin.

Stable cell lines of HuH3.1 FLAG:HA were made by transfecting 293TR cells with 1 µg of pcdna5 FRT huH3.1FLAG:HA, and 9 ug of POG44 (Life Technologies, V6005-20), using the Nucleofector kit according to the manufacturers instructions (Lonza Basel Switzerland, V4XC-2012). One day post transfection, cells were washed with fresh medium. Two days post transfection polyclonal stable cell lines were selected by maintaining cells in 400 µg/ml hygromycin. Stable cell lines expressing Aurora-A:FLAG, Aurora-A KD:FLAG and CapH:tGFP were made by transfecting 293TR cells (and desired H3 mutant cell lines) with 1 µg of plasmid, using lipofectamine 2000 according to the manufacturer's instructions. Two days post-transfection, stable cell lines were selected by maintaining cells in media containing 800 µg/ml G418.

## Tissue culture siRNA transfection

Cells were plated in a six-well dish and were grown to 50% to 60% confluence. For siRNA inhibition studies, the cells were co-transfected with 0.5 µg plasmid pBos H2B:RFP and siGENOME Human AURKA siRNA (Thermo Scientific Lafayette, CO USA, D-003545-05-0005) or ON-TARGET plus non-targeting siRNA #1 (Thermo Scientific, D-001810-01-05) (at a final concentration of 100 nM) in the presence of Lipofectamine 2000 reagent (Life Technologies, 11668019), as per the manufacturer's instructions. The cells were harvested at 72 hr post transfection for protein extraction and immuno-fluorescence analysis.

## Tissue culture shRNA transfections

For shRNA knockdown studies, three different shRNA constructs (pGipz) were purchased from MD Anderson's shRNA core. The target sequences of TPX2 shRNA are (1) TTAGCAGTGGAATCGAGTG; (2) AACAGGTTAATATCATCCT; (3) ATCTTGATGAGCACTGCCT. Cells were plated in six-well plates with CELL-TAK (BD Biosciences San Jose, CA USA, 354240) were grown to 50% to 60% confluence, and were cotransfected with all three TPX2 target sequences in the presence of Lipofectamine 2000 reagent (Life Technologies), as per the manufacturer's instructions. After transfection, the cells were split at 72 hr and 1 µg/ml puromycin was added. After 5 days the cells were collected for protein extraction and immunofluorescence analysis.

## *C. elegans* and RNAi mediated interference

Wild type N2 Bristol *C. elegans* were grown and maintained at 20°C as described (*Brenner, 1974*). The feeding method of RNAi delivery was used to deplete CENP-A/HCP-3, as previously described by Timmons and Fire (*Timmons and Fire, 1998*). RNAi plasmids for CENP-A/*hcp-3* were obtained from the Geneservice Ltd. *C. elegans* feeding library (*Kamath and Ahringer, 2003*). *E. coli* HT115 (DE3) bacteria was transformed with the control or CENP-A/HCP-3 RNAi plasmids. 1 ml LB + 100 µg/µl ampicillin liquid culture was inoculated with a single colony of HT115 bacterial transformation and grown overnight at 37°C. The following day these cultures were expanded into 50 ml LB/amp using a 1:100 dilution and grown for six hours at 37°C. After six hours, 200 µl were spread onto single nematode growth (NG) plates supplemented with 20% β-lactose and placed at 25°C for 72 hr. Subsequently, the plates were seeded with L4-stage hermaphrodites and incubated at 25°C for 24 hr (*Arur et al., 2009*). The L4440 RNAi vector was used as an RNAi control.

## Chromosome attachment error correction assay and drug treatments

We used an Eg5 inhibitor, Monastrol, to induce a monopolar spindle and kinetochore-microtubule attachment errors (*Sanhaji et al., 2010*). For the chromosome attachment error correction assays (monastrol-release experiments), cells were split into a 6 well dish at least 24 hr prior to treatment. Cells at 75% confluency were treated for 4 hr with monastrol (100 μM, Enzo Life Sciences, Farmingdale NY USA, BML-GR322-0005) and washed and released into fresh medium containing MG132 (20 μM, Calbiochem, Billerica, MA USA, 474790-1MG, in ETOH) for 2 hr and cells collected for immunofluorescence. All inhibitors were used at the listed concentrations MG132 (20 μM in ETOH), RO-3306 (9 μM, Enzo Life Sciences, ALX-270-463-M001, in DMSO), Nocodazole (100 mg/ml, Sigma, M1404, in ETOH), Colcemid (0.01 μg/mL, Roche 10295892001), PLK-1 inhibitor BI-2536 (100 nM, Selleck chemicals, Houston, TX USA, S1109, in DMSO), Caffeine (80 nM, Sigma C0750, in DMEM), Aurora-B inhibitor ZM447439 (2 μM, Tocris Biosciences, S1103, in DMSO), Calyculin A (50 nM, Tocris Biosciences, in EtOH), Aurora-B inhibitor Hesperidin (1 μM Selleck chemicals S2309, in DMSO), Aurora-A inhibitor VX-680 (1 μM, Selleck chemicals, S1048, in DMSO), Aurora-A inhibitor MLN 8237 (1 μM, Selleck Chem, S1133, in DMSO), Topoisomerase II inhibitor ICRF 193 (10 μM, Sigma, U4659, in DMSO).

## Tissue culture immunofluorescence

Immunofluorescence of metaphase chromosome spreads was prepared by cytospin following the pre-extraction method as described previously (*Ono et al., 2003*). Immunofluorescence of adherent cells were grown on poly-D-lysine coated coverslips (BD Biosciences, 354086) and harvested prior to reaching 80% confluency. Coverslips were washed in 1× PBS and fixed in 4% paraformaldehyde/1 x PBS for 10 min at room temperature (Electron Microscopy Sciences Hatfield, PA USA, 15710). Coverslips were washed in 1 x PBS and then permeabilized with 1 x PBS + 0.1% Triton X-100 at RT for 10 min. Coverslips were then washed in 1 x PBS and blocked in 3% BSA/1× PBS for 1 hr. Primary antibodies were diluted into 3% BSA/1 x PBS and incubated overnight at 4°C. Coverslips were washed 3 times 1× PBS for 15 min prior to adding secondary antibodies. Coverslips were washed 3 times in 1 x PBS for 15 min and mounted onto glass slides with ProLong Gold Antifade mounting reagent containing DAPI (Life Technologies, Grand Island, NY, USA, Cat# P36931). Immunofluorescence images were acquired as described below.

## *C. elegans* immunostaining

Embryos from adult hermaphrodites were picked into 10 μl egg buffer on a Poly-L-Lysine coated glass slide (Sigma, St Louis, MO P0425). To release the embryos, a coverslip was placed over the animals and gentle pressure was applied. The slides were subsequently placed on an aluminum plate over dry ice for 1 hr. To crack the embryo's cuticle and aid its permeabilization, coverslips were quickly snapped off. Slides were fixed in -20°C methanol for 20 min, followed by sequential rehydrations: 80:20, 50:50, and 20:80 methanol to 1x PBS with 0.1% Tween (PBST). After hydration, samples were blocked in 1X PBST with 1% BSA for 1 hr at room temperature and then incubated overnight in primary antibody diluted in PBST at 4°C. Primary antibodies used were anti-tubulin (1:2000, Sigma), and H3 T118ph (1:1000). Samples were then washed with PBST and secondary antibodies were applied for 2 hr at room temperature. Secondary antibodies used were: Alexa Fluor 488 goat anti-mouse IgG and Alexa Fluor 594 goat anti-rabbit (both at 1:1000) (Invitrogen Molecular Probes, Eugene, OR). After incubation with the secondary antibodies the samples were washed with PBST and mounted using ProLong Gold Antifade ProLong with DAPI. Immunofluorescence images were acquired as described below.

## Mitotic chromosome spreads

Cells were collected by mitotic shake off. Media was removed and the cells were pelleted at 1000 rpm for 5 min. All but 1 ml of media was removed and gently used to resuspend cells. Cells were swollen in 10 ml of hypotonic solution (46.5 mM KCl/8.5 mM NaCitrate) and incubated for 20 min at 37°C. Fresh Carnoy's fixative (3:1 methanol:acetic acid) was added to hypotonic buffer at 10% (v/v). Subsequent to centrifugation cells, were fixed 3 times with 10 mls Carnoy's fixative for 10 min at RT followed by pelleting the cells at 1000 rpm for 5 min. Pellets were than resuspended in a small volume of Carnoy's fixative, dropped onto positively charged slides (Fisher scientific, Ashville, NC

USA,12-550-15) air-dried, and stained with 1 mg/ml DAPI solution diluted 1:15,000. Slides were mounted with ProLong Gold Antifade mounting reagent containing DAPI. Immunofluorescence images were acquired as described below. To stain heterochromain, chromosome spreads were treated as in (Hirota et al., 2004) except 0.08 mg/ml netropsin was used instead of distamycin.

## SEM

We followed published methods (Lai et al., 2011). Chromosome spreads were prepared as described above except the chromosomes were dropped onto poly-D-lysine coated coverslips (BD Biocoat, 354086) in a 37°C room with minimal drying. The coverslips were flipped onto a larger coverslip with 1 drop of 45% acetic acid and the large coverslip was placed on dry ice for 15 min. The chromosome spreads were then fixed in 2.5% glutaraldehyde / 1 x PBS overnight at 4°C. The fixed samples were than washed with distilled water for 5 min, 10 min, and 15 min, then dehydrated with a graded series of increasing concentrations of ethanol (5 min in 70%, 10 min in 90% and 15 min in 100%). The samples were then chemically dried in a graded series of increasing concentrations of hexamethyldisilazane (HMDS, Electron Microscopy Services) 2:1 (100% EtOH:HMDS), 1:1 (100% EtOH: 100% HMDS), then 1:2 (100% EtOH: HMDS), then 3 changes in pure HMDS where all steps were for 5 min each. Then the samples were air dried overnight. Samples were mounted onto an aluminum specimen mount (Ted pella, INC.) by carbon conductive double-stick tape (Ted Pella. Inc., Redding, CA). The samples were then coated under vacuum using a sputter system (208HR, Cressington Scientific Instruments, England) with platinum alloy for a thickness of 30 nm. Samples were examined in a Nova NanoSEM 230 scanning electron microscope (FEI, Hillsboro, Oregon) at an accelerating voltage of 10 kV.

## Indirect immunofluorescence of chromosome spreads

In general, to produce chromosome spreads, HeLa mitotic cells obtained by mitotic shake off were incubated in pre-warmed hypotonic buffer (46.5 mM KCl/8.5 mM NaCitrate) at 37°C for 8–10 min. 293TR mitotic cells obtained by selective detachment were incubated in pre-warmed hypotonic buffer #5 (10 mM Tris-HCl pH7.4, 40 mM glycerol, 20 mM NaCl, 1.0 mM $CaCl_2$, 0.5 mM $MgCl_2$). After attachment to Poly-D-lysine glass coverslips by Cytospin at 1000 rpm for 2 min, chromosome spreads were pre-extracted with 0.1% Triton X-100/1 x PBS for 2 min and were than fixed with 2% PFA/1 x PBS at RT for 10 min. Cells were extracted with 0.1% Triton X-100/PBS for 10 min. Blocking occurred in 1 x PBS, 3% BSA, and 0.1% Triton X-100, for 30 min at room temperature. Once blocking was complete, the immunofluorescence protocol was followed as described above.

## Extended chromatin fibers

Cells were arrested with colcemid and the chromatin fibers were generated as described elsewhere (Dunleavy et al., 2011). Briefly, chromatin fibers from human cells were prepared by incubating 6–8 x $10^4$ cell/ml in prewarmed hypotonic buffer at 37°C for 10 min. HeLa cells used hypotonic buffer 46.5 mM KCl/8.5 mM Na Citrate and for 293TR used the buffer was 10 mM Tris-HCl pH 7.4, 40 mM glycerol, 20 mM NaCl, 1.0 mM $CaCl_2$, 0.5 mM $MgCl_2$. Cells were centrifuged onto charged microscope slides (Fisher Scientific, 2-550-15) and lysed for 14 min in salt detergent buffer supplemented with urea (10 mM Tris HCl pH 7.5, 1% Triton X-100, 500 mM NaCl, and 500 mM urea) before slowly aspirating the lysis buffer by vacuum and fixing in 2% PFA/1 x PBS. Slides were incubated in $1\times$ PBST ($1\times$ PBS + 0.1% Triton X-100) and blocked in 1 x PBS, 1% BSA, 0.1% Triton X-100, for 30 min at room temperature. Once blocking was complete, the immunofluorescence protocol was followed as described above.

## Isolation of pellet and supernatant fractions

Two D150 plates, at 80% confluency, were collected by mitotic shake off. Cells were pelleted and washed in TB buffer (20 mM Hepes, pH 7.3, 110 nM K-acetate, 5 mM Na-acetate, 2 mM Mg-acetate, 1 mM EGTA, 2 mM DTT, and a protease inhibitor cocktail (Roche, Complete-mini, cat#1187350001). All steps were done at 4°C. NP40 extraction of detergent soluble proteins was performed by treatment with 0.1% NP40 for 5 min, followed by centrifugation at 3000 rpm for 3 min to separate the non-chromatin supernatant and chromatin pellet fractions. The pellet fractions were subsequently digested with 20 µg/ml DNaseI (Worthington Biochemical Corporation, Lakewood, NJ USA,

LS006342) for 10 min at 37°C. Total, supernatant (non-chromatin), and pellet (chromatin) fractions were resolved by SDS-PAGE and analyzed by western blotting.

## Differential salt solubility

The method was adapted from (*Henikoff et al., 2008*) with the differences detailed below. Five million cells were pelleted during the nuclei extraction on ice samples and were divided into 5 tubes. The Nuclei were washed in NIM buffer (0.25 M sucrose, 25 mM KCl, 5 mM MgCl$_2$, 10 mM Tris-HCL pH 7.4). Pelleted at 300 rpm for 5 min. The nuclei were resuspened in 5 different extraction buffers at increasing salt concentration (0, 600, 900, 1200, and 1500 mM NaCl) Incubated on ice for 10 min. The soluble and pellet fractions were collected by centrifugation at 13,000 rpm for 10 min. 5xSDS was added to the soluble fractions and boiled at 100°C for 5 min. The pellet fractions were resuspended in 250 µl Laemmli buffer and the Whole Cell Extract protocol was followed (as detailed below).

## Whole cell extracts

Approximately $2 \times 10^6$ cells were lysed with 200 µl Laemmli buffer (4% SDS, 20% glycerol, and 120 mM Tris pH 6.8). Cells were subsequently vortexed for 30 s and then boiled at 100°C for 5 min. After briefly cooling, samples were vortexed for 30 s and sonicated for 10 s at 20% power. Lastly, 5 x SDS buffer was added to samples to obtain a 1x final concentration and samples were boiled at 100°C for 5 min.

## Immunoprecipitation

Whole cell extracts were prepared using RIPA buffer (150 mM NaCl, 1% NP40, 0.5% sodium deoxycholate, 0.1% SDS, 50 mM Tris, pH 8, 10 mM NaF, 0.4 mM EDTA, 10% glycerol and protease inhibitors) supplemented with 10x phostop (Roche 04906845001), and 25x Protease inhibitor (Roche 04693132001). The pre-blocked protein-A–Dynabeads (Thermo-Fischer, 10001D) was then incubated with whole-cell extracts overnight in 4°C. The antibody was added for 4 hr the next day. Following extensive washes, the bead-bound protein complexes were analyzed by western blotting using H3 C-term antibody.

## Western blot analysis

Samples were resolved by 15% SDS-PAGE and transferred to nitrocellulose according to standard procedures. For HRP detection, following transfer the membranes were blocked in 5% non-fat milk (w/v) in 1× TBST for 1 hr. The blots were probed with primary antibodies at room temperature for 1 hr or overnight at 4°C. Blots were washed and incubated in secondary antibodies at room temperature for 1 hr. ECL detection was either by Amersham ECL Western Blotting Detection Reagents (GE, Pittsburgh, PA USA, RPN2106) or Immobilon Western Chemiluminescent HRP Substrate (Millipore, WBKLS0500). Alpha viewer was used to analyze and quantitate bands (Proteinsimple, Santa Clara, CA, USA). For LICOR Odyssey detection the transfer blots were blocked in Sea Block buffer (Thermo Scientific, Cat#37527) in 1 x PBS for 1 hr. Blots were incubated with primary and secondary antibodies as described above. An Odyssey imager was used to analyze and quantitate bands.

## Peptide dot blots

Lyophilized peptides were rehydrated in 1 x PBS at a 10 µM concentration. The peptides were serially diluted to the indicated concentrations and dotted out onto activated PVDF membrane. The membrane was air-dried and then stained with amido black to verify the presence of the peptides. The membranes were washed in PBS and then blocked in 3% BSA/1 x PBS. The blots were incubated in primary antibodies overnight at 4°C. The blots were washed and probed with HRP conjugated secondary antibodies at room temperature for 1 hr. Alpha viewer was used to analyze and quantitate bands (Proteinsimple).

## Peptide competition assay

Biotinylated peptides (4 µg total) were incubated with 400 µl H3 T118ph antibody for 45 min at room temperature (RT). Samples were centrifuged for 15 min at 4°C at 13 k rpm. 300 µl of each supernatant was used for indirect immunofluorescence as described above.

## DNase 1 fragmentation assay

Approximately $2 \times 10^6$ cells were lysed with 2 mL of lysis buffer (50 mM Tris-HCl pH 7.9, 100 mM KCl, 5 mM MgCl2, 0.05% v/v saponin, 50% v/v glycerol, 0.5M DTT, 10x phostop (Roche 04906845001), and 25x Protease inhibitor (Roche 04693132001) of asynchronous or synchronized cells (synchronized for 6 hr in nocodazole (100 mg/ml, in ETOH)). Cells were incubated in lysis buffer for 3 min on ice and vortexed every minute. Samples were centrifuged for 10 min at 4°C at 1000 x g. Nuclei were subsequently digested for increasing times at 37°C with 5U DNase I (Worthington Biochemical Corporation LS006342) in TB buffer (20 mM Hepes, pH 7.3, 110 nM Potassium-acetate, 5 mM Sodium-acetate, 2 mM Magnesium-acetate, 1 mM EGTA, 2 mM DTT and a protease inhibitor cocktail (Roche, Complete-mini, cat#1187350001)). Fragmented DNA was purified and analyzed by agarose gel electrophoresis followed by Sybr Gold (Life Technologies, S-11494) staining for visualization with a FluorChem E FE05000 (Protein simple, San Jose, CA). Plot profiles were obtained with ImageJ software.

## Live cell imaging

A four well chamber was coated with BD Bio TAK according to the manufacturer's instructions. Approximately 24 hr prior to live cell imaging, HEK293 cells were transfected using the Nucleofector kit according to the manufacturers instructions (Lonza,V4XC-2012) with 0.5 μg plasmid CMV:H3.2 YFP wild type or T118 mutant. The transfected cells were plated at 50,000 cells per well and grown in a humidified chamber for 24 hr. At the time of imaging, the cells were placed in a prewarmed Oko Full Enclosure incubator at 37°C with 5% $CO_2$. Cells were imaged using a 3i Marianas Spinning Disk Confocal equipped with an Evolve 10 MHz Digital Monochrome Camera (Photometrics, Tuscon, AZ USA) and images were taken every 5 min for 16 hr and driven by Slidebook 5.5 software (a 63 x 1.49 NA Plan Apo oil immersion objective). Three Z-sections were acquired for each cell. The start of cytokinesis was defined when H3:YFP chromatin decondensed after anaphase. The end of cytokinesis was determined by the physical separation of the cytoplasmic membrane.

## Acquisition of images

The majority of images were acquired on a 3i Marianas Spinning Disk Confocal equipped with a coolSNAP HQ2 CCD Camera. Slidebook 5.5 software was used with a 63 x 1.49NA Plan Apo oil immersion objective and Z sections were acquired at 0.2 um steps. Intensity measurements were calculated with Slidebook 5.5 software. To measure inter-kinetochore distance, the center intensity of foci was determined by Imaris Bitplane software.

Some immunofluorescence images were acquired on a Nikon 2000U inverted microscope equipped with a Photometrics Coolsnap HQ camera. Metamorph software was used with a 60x 1.49NA Plan Apo oil immersion objective and Z sections were acquired at 0.2 μm steps.

## EMSA

Histone octamers unmodified and modified at H3T118ph were purified according to the method North et. al. (*North et al., 2011*). The Condensin I complex was purified from $5 \times 10^6$ CAP-H-GFP-SBP, SMC2-SBP and GFP-SBP mitotic cells using the method by Kim et. al. (*Kim et al., 2010b*). After purification, proteins were eluted in SEB (50 mM Tris pH7.4, 250 mM NaCl, 0.5% NP-40, 0.1% Deoxycholate and 4 mM Biotin). 10 ml samples were subjected to NuPAGE SDS-PAGE and protein and evaluated by silver staining. The EMSA was performed as described previously (*Kimura et al., 1997*).

## Generation of flies with H3 T118 mutations

The following genotypes were used in this study:

  *yw; Df(2L)HisC/ CyO, P{ActGFP}JMR1; 6xHisGUVK33,27/ TM6B*
  *yw; Df(2L)HisC/ CyO, P{ActGFP}JMR1; 6xHisGU VK33,27 H3T118A/ TM6B*
  *yw; Df(2L)HisC/ CyO, P{ActGFP}JMR1; 6xHisGU VK33,27 H3T118E/ TM6B*
  *yw; Df(2L)HisC/ CyO, P{ActGFP}JMR1; 6xHisGU VK33,27 H3T118I/ TM6B*

We constructed *6xHisGU VK33,27* and *6xHisGU VK33,27 H3T118A, E* and *I* chromosomes essentially as previously described (*Gunesdogan et al., 2010*) with the following changes: ΦC31attB3xHisGU. H3T118A, ΦC31attB3xHisGU.H3T118E, and ΦC31attB3xHisGU.H3T118I plasmids (further referred

to collectively as H3T118A/E/I) were generated by replacing the EcoR1/Sac1 fragment in pENTR221-HisGU with a synthetic fragment (Integrated DNA Technologies, Inc., Iowa, USA) containing an ACC into GCC codon exchange leading to the H3 T118A mutation, an ACC into GAG codon exchange leading to the H3 T118E mutation, or an ACC into AUC codon exchange leading to the H3 T118I mutation. The pENTRL4R1-HisGU.H3T118A/E/I and pENTRR2L3-HisGU.H3T118A/E/I entry vectors were generated by moving the Acc65I/AgeI fragment from the pENTR221-HisGU.H3T118A/E/I mutant vectors to the pENTRL4R1 and the pENTRR2L3 vectors. Recombination of pENTR221-HisGU.H3T118A/E/I, pENTRL4R1-HisGU.H3T118A/E/I and pENTRR2L3-HisGU.H3T118A/E/I with pDESTR3R4-ΦC31attB resulted in the ΦC31attB3xHisGU.H3T118A/E/I transgenic constructs. We utilized ΦC31-mediated transgenesis to integrate these constructs, as well as ΦC31attB3xHisGU, site specifically into the *Drosophila* genome using the landing sites *VK27* and *VK33* (*Venken et al., 2006*). Homozygous viable insertions from each site were recombined to generate *6xHisGU* $^{VK33,27}$ and *6xHisGU* $^{VK33,27}$ *H3T118A, E* and *I* chromosomes and crossed into the *Df(2L)His$^C$* mutant background (*Gunesdogan et al., 2010*). *Df(2L)His$^C$* was kept heterozygous over *CyO, P{ActGFP} JMR1 to* identify mutant embryos lacking green fluorescent protein expression, and the viability of *12xHisGU* transgene containing mutant and wild type animals was assessed. Wild type controls were either non-mutant sibling embryos (internal control) or embryos which contain *12xHisGU* (WT control), which both survive to adult viability.

## Acknowledgements

We acknowledge Ted Salmon for providing antibodies and Walter Hittelman, Sen Subrata, and Zhenkun Lou for plasmids. We are grateful to Rick Fishel for discussions at the early stages of this project. We thank Henry Adams for his assistance with microscopy and analysis. We thank Alf Herzig for the plasmids and fly stocks. We are grateful to Hugo Bellen, Yuchun He and Hongling Pan for *Drosophila* embryo injections. We thank Alan J Tackett and Stephanie Byrum for mass-spec analyses, Kenneth Dunner Jr for help with the SEM, Jinhua Gu (SEM/AFM core at Houston Methodist Hospital Research Institute) for assistance. We thank Ngoc H. Bui for technical assistance and Jason Ford, Jessica Orbeta, and Tokiko Furuta for help with *C. elegans* experiments. We thank Brandee Price and Ja-Hwan Seol for comments on the manuscript. We thank Brad Cairns for his support during the review process. This work was supported by NIH grant GM64475.

## Additional information

### Competing interests

JKT: Reviewing editor, *eLife.* The other authors declare that no competing interests exist.

### Funding

| Funder | Grant reference number | Author |
| --- | --- | --- |
| National Institutes of Health | RO1GM64475 | Jessica K Tyler |
| National Institutes of Health | RO1 CA95641 | Jessica K Tyler |

The funders had no role in study design, data collection and interpretation, or the decision to submit the work for publication.

### Author contributions

CLW, HKG, Conception and design, Acquisition of data, Analysis and interpretation of data, Drafting or revising the article; RH, MDG, MBF, TZ, ZC, Acquisition of data; DFH, JJO, Contributed unpublished essential data or reagents; MGP, Analysis and interpretation of data, Contributed unpublished essential data or reagents; JS, Conception and design, Analysis and interpretation of data; JKT, Analysis and interpretation of data, Drafting or revising the article

### Author ORCIDs

Jessica K Tyler, http://orcid.org/0000-0001-9765-1659

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
