## [Decision Letter]

Thank you for submitting your work entitled "Aurora-A mediated histone H3 phosphorylation of threonine 118 controls condensin I and cohesin occupancy in mitosis" for consideration by *eLife*. Your article has been reviewed by two peer reviewers, and the evaluation has been overseen by a Reviewing Editor and James Manley as the Senior Editor.

The reviewers have discussed the reviews with one another and the Reviewing Editor has drafted this decision to help you prepare a revised submission.

Summary:

This manuscript by the Tyler lab studies the role of H3T118 phosphorylation in mitosis. Histone H3-T118 is part of a network of amino acids that position a key arginine residue (H4-R45) for interaction with nucleosomal DNA near the dyad axis. Mutations in H3T118 were first identified by genetic screens in budding yeast, and subsequent biochemical studies showed that this residue was phosphorylated in cells and that substitutions in T118 or its phosphorylation destabilized nucleosomal structure. The present study is of general interest because the physiological function of H3T118-Phos is unknown.

H3T118-phos function is examined in human cells, *Drosophila*, and *C. elegans*. Using an antibody that recognizes T118-phos, the authors find that this mark is cell cycle-regulated, appearing in early mitosis where it marks the centrosomes, centromeres, and chromosome arms. Using an in vitro kinase screen, they identify Aurora A as the likely kinase. Knockdown of Aurora A appears to eliminate T118-phos from mitotic chromosomes. Based largely on results in HeLa cell lines that express exogenous histones harboring substitutions of T118 (T118A, T118E, and T118I), the authors conclude that T118-phos is important for proper chromosome alignment, and that it controls chromatin association of cohesin and condensin. Although the mutant histones are only expressed at about 10% of the level of endogenous histones, the T118E and T118I derivatives indeed caused a range of mitotic defects: lagging chromosomes, mitotic delays, loss of cohesion and condensin, and altered mitotic chromosome structure. Confirming the authors’ model, similar defects are observed after over-expression of Aurora A.

The findings presented in the paper are of general interest, and the experiments are overall of good standard. However, there are a number of reservations, several of which will have to be addressed experimentally. We would, therefore, be interested in a revised manuscript that resolves the points outlined below.

Essential revisions:

1) The most critical issue is to include a better characterization of the fly embryos with the mutated H3T118. This is important because much of this manuscript focuses on data where the ectopic H3T118 mutants are expressed at no more that 10% of the endogenous H3. The low percentage of mutant H3 might well explain why H3T118A has no clear phenotype. Nevertheless, to conclude that T118 phosphorylation is important for mitosis, this is a critical "loss-of function" mutant. This conundrum could be remedied by the studies in *Drosophila*, where flies are engineered to express H3 derivatives as the sole source of H3. These flies die during embryogenesis as soon as the maternal histone deposits have been depleted. These embryos should be analyzed for a mitotic block or chromosome segregation defects. Does the H3T118A mutant cause segregation errors? And, do the H3T118E or H3T118I mutants display a different phenotype? These experiments may also ameliorate some of the other concerns raised below. Alternatively, the *Drosophila* experiments may indicate that the histone substitutions lead to general loss of viability, not necessarily a specific mitotic defect. This would be similar to the case in budding yeast.

2) There are some concerns with respect to the antibody characterization presented (Figure 1). (1) Controls should include a T118 peptide that is not phosphorylated. (2) The exposure in Figure 1 is not sufficient to evaluate whether other proteins are recognized. Likewise, the complete gel should be shown in Figure 1. (3) The immunoprecipitation data is not convincing.

3) The authors’ results suggest that H3T118 is phosphorylated by Aurora A, and that this is important for error correction. However, it would be prudent to ensure that this phosphorylation is not dependent on Aurora B in vivo. One could include a low dose of the Aurora B inhibitor ZM447439 in Figure 3. Error correction is known to be highly dependent on Aurora B, and the consensus sites are the same. It would then be necessary to verify that classical Aurora B targets are not phosphorylated in this case, while H3T118 is.

4) It is not clear why wild type histone H3 (and bulk H3) is removed from chromatin at only 90 mM NaCl (Figure 5). Removal of histone H3 from a nucleosome requires Na+ concentrations > 1M. A concern is that the authors are looking at a pool of H3 that is weakly bound to DNA, but not nucleosomal. Related to this, the authors need to demonstrate that the exogenous histone substitution alleles are actually incorporated into nucleosomes (Figure 3). Fractionation into the chromatin pellet might only indicate nuclear import and nonspecific DNA association. A simple MNase digestion of nuclear chromatin, followed by a glycerol gradient fractionation step would test this.

5) In the second paragraph of the subsection “H3 T118I and T118E cause premature removal of cohesin from DNA”. The logic here is unclear. If H3T118-phos directly promotes cohesin removal, then one would predict no co-localization, as cohesin would be removed wherever there was T118-phos. The lack of T118-phos on heterochromatin seems inconsistent with the centromeric localization of this mark. Why would T118 phosphorylation need to promote cohesin dissociation at centromeres? Surely, Separase cleaves cohesin at anaphase onset?

6) Based on the presented data (Figure 6), one cannot conclude that T118 affects cohesion in a Shugoshin independent manner. Different sets of hands often get different results regarding partially separate vs. fully separate chromatids. If the authors wish to stress this point, they should include Sgo1 immune fluorescence to assess whether Sgo1 localisation is affected by the T118 mutations. It is likely though that Sgo1 is decreased, as cohesin is required for efficient Sgo1 docking at centromeres. Therefore, if Sgo1 is lost from centromeres, one cannot draw conclusions regarding the loss of cohesion. Considering that the loss of cohesion depends on the kinases Plk1 and Aurora B (Figure 6), it could be argued that the phenotype is caused by loss of Sgo1-mediated protection against these kinases.

---

## [Author Response]

*[…] The findings presented in the paper are of general interest, and the experiments are overall of good standard. However, there are a number of reservations, several of which will have to be addressed experimentally. We would, therefore, be interested in a revised manuscript that resolves the points outlined below. Essential revisions: 1) The most critical issue is to include a better characterization of the fly embryos with the mutated H3T118. This is important because much of this manuscript focuses on data where the ectopic H3T118 mutants are expressed at no more that 10% of the endogenous H3. The low percentage of mutant H3 might well explain why H3T118A has no clear phenotype. Nevertheless, to conclude that T118 phosphorylation is important for mitosis, this is a critical "loss-of function" mutant. This conundrum could be remedied by the studies in Drosophila, where flies are engineered to express H3 derivatives as the sole source of H3. These flies die during embryogenesis as soon as the maternal histone deposits have been depleted. These embryos should be analyzed for a mitotic block or chromosome segregation defects. Does the H3T118A mutant cause segregation errors? And, do the H3T118E or H3T118I mutants display a different phenotype? These experiments may also ameliorate some of the other concerns raised below. Alternatively, the Drosophila experiments may indicate that the histone substitutions lead to general loss of viability, not necessarily a specific mitotic defect. This would be similar to the case in budding yeast.*

To answer the reviewers request for information on the effect of T118A in *Drosophila*, we have performed an extensive analysis of the H3 T118A mutants and have now added to the Discussion section “*Drosophila* cell clones expressing only H3 T118A have significant defects in cell growth and viability (Graves et al., submitted).” This is consistent with our proposed role of T118p in promoting timely removal of cohesin and condensin. Furthermore, the defects in cell growth for T118E and T118I are even more pronounced than for T118A (Graves et al., submitted), consistent with excess T118p driving premature cohesin and condensin loss. We were unable to directly examine whether *Drosophila* embryos expressing T118A have mitotic defects because we found that in our hands the transgenes carrying the 12 wild type histone genes were not able to fully rescue the mitotic defect previously observed in Cycle 15 embryos lacking endogenous histones (EMBO Reports and *eLife*) although they do rescue viability. In addition, unfortunately the *Drosophila* chromosomes in imaginal discs are too small to examine for mitotic defects in the clones of cells expressing T118A that we generated (Graves et al., submitted). Regardless, we have removed the conclusion that T118p is important for mitosis from the current text.

*2) There are some concerns with respect to the antibody characterization presented (Figure 1). (1) Controls should include a T118 peptide that is not phosphorylated. (2) The exposure in Figure 1 is not sufficient to evaluate whether other proteins are recognized. Likewise, the complete gel should be shown in Figure 1. (3) The immunoprecipitation data is not convincing.*

Thank you for helping us realize that our labels are not clear. For both the dot blot and the peptide competition assay we utilized an unmodified peptide. To make this more clear we have replaced “H3 115-125aa” with “unM T118” in the figure and updated the figure legends to explain that unM T118 is the unmodified 10 amino acid peptide spanning H3 115-125aa. As requested, we have now provided better images to address points (2), and (3). Please see Figure 1 and the complete gel of the western blots in Figure 1—figure supplement 1.

*3) The authors’ results suggest that H3T118 is phosphorylated by Aurora A, and that this is important for error correction. However, it would be prudent to ensure that this phosphorylation is not dependent on Aurora B in vivo. One could include a low dose of the Aurora B inhibitor ZM447439 in Figure 3. Error correction is known to be highly dependent on Aurora B, and the consensus sites are the same. It would then be necessary to verify that classical Aurora B targets are not phosphorylated in this case, while H3T118 is.*

We had previously shown that knock down of TPX2 and Aurora A blocked H3 T118, while overexpression of Aurora A leads to more H3 T118p, and that Aurora A can phosphorylate H3 T118ph in vitro, suggesting that Aurora A is directly involved in the phosphorylation of H3 T118. However, as requested we treated cells with a variety of concentrations of ZM447439. As expected, we found that treatment of ZM447439, which decreased levels of the Aurora B target H3 S10 (Figure 9), additionally inhibited Aurora A function (Figure 9), as evidenced by the loss of Aurora-A T288 phosphorylation. This is likely due to the fact that inhibiting Aurora B will inhibit PLK-1 [1] which in turn prevents TPX2 phosphorylation and that in turn prevents Aurora A localization to chromatin [2]. Therefore, we would expect to lose H3 T118ph when we inhibit Aurora B with ZM447439 (or by any means) because of this indirect effect on the activity of Aurora A. This is exactly what we saw. As such, the use of Aurora B inhibition with ZM447439 is not very informative.

Author response image 1.**DOI:**
http://dx.doi.org/10.7554/eLife.11402.027

*4) It is not clear why wild type histone H3 (and bulk H3) is removed from chromatin at only 90 mM NaCl (Figure 5). Removal of histone H3 from a nucleosome requires Na+ concentrations > 1M. A concern is that the authors are looking at a pool of H3 that is weakly bound to DNA, but not nucleosomal. Related to this, the authors need to demonstrate that the exogenous histone substitution alleles are actually incorporated into nucleosomes (Figure 3). Fractionation into the chromatin pellet might only indicate nuclear import and nonspecific DNA association. A simple MNase digestion of nuclear chromatin, followed by a glycerol gradient fractionation step would test this.*

We thank the reviewers for this comment, which alerted us to the fact that we labeled the gel wrong. The numbers were meant to be 600nm and 900nm NaCl and not 60nm and 90nm and we have now corrected this error. We have also added better experiment details in the Materials and methods section.

*5) In the second paragraph of the subsection “H3 T118I and T118E cause premature removal of cohesin from DNA”. The logic here is unclear. If H3T118-phos directly promotes cohesin removal, then one would predict no co-localization, as cohesin would be removed wherever there was T118-phos. The lack of T118-phos on heterochromatin seems inconsistent with the centromeric localization of this mark. Why would T118 phosphorylation need to promote cohesin dissociation at centromeres? Surely, Separase cleaves cohesin at anaphase onset?*

The reviewers are correct, and based on a later comment where you suggest we make an effort to streamline our Results/excessive data, we decided to remove this section of the paper and the data related to it, because it does not improve the overall scope of the research article.

6) Based on the presented data (Figure 6), one cannot conclude that T118 affects cohesion in a Shugoshin independent manner. Different sets of hands often get different results regarding partially separate vs. fully separate chromatids. If the authors wish to stress this point, they should include Sgo1 immune fluorescence to assess whether Sgo1 localisation is affected by the T118 mutations. It is likely though that Sgo1 is decreased, as cohesin is required for efficient Sgo1 docking at centromeres. Therefore, if Sgo1 is lost from centromeres, one cannot draw conclusions regarding the loss of cohesion. Considering that the loss of cohesion depends on the kinases Plk1 and Aurora B (Figure 6), it could be argued that the phenotype is caused by loss of Sgo1-mediated protection against these kinases.

The reviewers have a valid point and we have attempted to assess Sgo-1 levels by immunofluorescence in the T118 mutants. Unfortunately the analyses were not conclusive due to the low quality of the commercial antibodies that are available. Accordingly, we have removed the previous experiments that were deemed inconclusive and have removed the conclusion that T118 affects cohesion in a Shogoshin independent manner, and have noted the possibility in the Discussion, that the cohesion phenotype may be due to loss of Sgo-1 mediated protection against the PLK-1 and Aurora-B kinases.

References:

1) Archambault, V. and M. Carmena, Polo-like kinase-activating kinases: Aurora A, Aurora B and what else? Cell Cycle, 2012. 11(8): p. 1490-5.

2) De Luca, M., P. Lavia, and G. Guarguaglini, A functional interplay between Aurora-A, Plk1 and TPX2 at spindle poles: Plk1 controls centrosomal localization of Aurora-A and TPX2 spindle association. Cell Cycle, 2006. 5(3): p. 296-303.